# p38γ and p38δ modulate innate immune response by regulating MEF2D activation

**Alejandra Escós**[1†], **Ester Diaz-Mora**[1†], **Michael Pattison**[2], **Pilar Fajardo**[1], **Diego González-Romero**[1], **Ana Risco**[1], **José Martín-Gómez**[1], **Éric Bonneil**[3], **Nahum Sonenberg**[4,5], **Seyed Mehdi Jafarnejad**[6], **Juan José Sanz-Ezquerro**[7], **Steven C Ley**[2,8], **Ana Cuenda**[1]*

[1]Department of Immunology and Oncology, Centro Nacional de Biotecnología/CSIC (CNB-CSIC), Campus-UAM, Madrid, Spain; [2]The Francis Crick Institute, London, United Kingdom; [3]Institute for Research in Immunology and Cancer, Université de Montréal, Montreal, Canada; [4]Goodman Cancer Research Center, McGill University, Montreal, Canada; [5]Department of Biochemistry, McGill University, Montréal, United Kingdom; [6]Patrick G. Johnston Centre for Cancer Research, Queen's University Belfast, Belfast, United Kingdom; [7]Department of Molecular and Cellular Biology, CNB-CSIC, Madrid, Spain; [8]Institute of Immunity & Transplantation, University College London, London, United Kingdom

**\*For correspondence:**
acuenda@cnb.csic.es

†These authors contributed equally to this work

**Abstract** Evidence implicating p38γ and p38δ (p38γ/p38δ) in inflammation are mainly based on experiments using *Mapk12/Mapk13*-deficient (p38γ/δKO) mice, which show low levels of TPL2, the kinase upstream of MKK1–ERK1/2 in myeloid cells. This could obscure p38γ/p38δ roles, since TPL2 is essential for regulating inflammation. Here, we generated a *Mapk12*[D171A/D171A]/*Mapk13*[−/−] (p38γ/δKIKO) mouse, expressing kinase-inactive p38γ and lacking p38δ. This mouse exhibited normal TPL2 levels, making it an excellent tool to elucidate specific p38γ/p38δ functions. p38γ/δKIKO mice showed a reduced inflammatory response and less susceptibility to lipopolysaccharide (LPS)-induced septic shock and *Candida albicans* infection than wild-type (WT) mice. Gene expression analyses in LPS-activated wild-type and p38γ/δKIKO macrophages revealed that p38γ/p38δ-regulated numerous genes implicated in innate immune response. Additionally, phospho-proteomic analyses and in vitro kinase assays showed that the transcription factor myocyte enhancer factor-2D (MEF2D) was phosphorylated at Ser444 via p38γ/p38δ. Mutation of MEF2D Ser444 to the non-phosphorylatable residue Ala increased its transcriptional activity and the expression of *Nos2* and *Il1b* mRNA. These results suggest that p38γ/p38δ govern innate immune responses by regulating MEF2D phosphorylation and transcriptional activity.

## Editor's evaluation

The authors describe a new mouse model that can be used to examine p38γ/δ MAP kinase function. The data presented are solid and convincing. The authors show that p38γ/δ MAP kinase signaling contributes to macrophage responses to endotoxin. Moreover, the authors identify Ser444 as an inhibitory site of MEF2D phosphorylation by p38d.

## Introduction

The p38 mitogen-activated protein kinases (p38MAPK) are essential in cell adaptation to environmental changes, as well as in inflammatory cellular responses to pathogen infection (*Cuenda and Rousseau, 2007*; *Cuenda and Sanz-Ezquerro, 2017*). The stimulation of pattern-recognition

receptors, by pathogens or molecules from damaged cells, triggers the activation of p38MAPK and other signalling pathways that are critical to promote an innate immune response and production of inflammatory molecules (cytokines and other mediators) (*Gaestel et al., 2009*; *Arthur and Ley, 2013*). It is well established that among the four p38MAPK isoforms (p38α (*Mapk14*), p38β (*Mapk11*), p38γ (*Mapk12*), and p38δ (*Mapk13*)), p38α is an inflammation modulator and has been considered as a therapeutic target for inflammatory diseases and related pathologies (*Cuenda and Sanz-Ezquerro, 2017*; *Gaestel et al., 2009*; *Arthur and Ley, 2013*; *Han et al., 2020*). However, clinical trials using potent p38α inhibitors have failed (*Arthur and Ley, 2013*; *Cohen, 2009*).

p38γ and p38δ (p38γ/p38δ) are closely related kinases and have largely redundant functions, which makes it difficult to study the specific functions of these kinases using single knock-out mice. Recent studies have analysed mice lacking both p38γ and p38δ (p38γ/δKO) kinases to demonstrate the importance of these kinases in the innate immune response and inflammation. p38γ/p38δ promote inflammation in several disease models, including sepsis, candidiasis, colitis, dermatitis, liver steatosis, arthritis, and cancer associated with inflammation (*Cuenda and Sanz-Ezquerro, 2017*; *Alsina-Beauchamp et al., 2018*). Biochemical studies have indicated an important role for p38γ/p38δ MAPKs in myeloid cells (*Cuenda and Sanz-Ezquerro, 2017*; *Alsina-Beauchamp et al., 2018*; *Risco et al., 2012*). The production of cytokines and chemokines triggered by C-type lectin receptor (Dectin-1) and Toll-like receptor (TLR) stimulation is reduced in macrophages lacking p38γ/p38δ (*Alsina-Beauchamp et al., 2018*). Also, compound p38γ/p38δ deficiency substantially reduces *Map3k8* mRNA translation and TPL2 protein levels in macrophages (*Alsina-Beauchamp et al., 2018*; *Risco et al., 2012*; *Escós et al., 2022*). TPL2 is the key MAP3-kinase upstream of the MKK1–extracellular signal-regulated kinase 1/2 (ERK1/2) in myeloid cells in innate immune responses. p38γ/δKO macrophages show impaired TLR activation of ERK1/2 and reduced production of key TPL2-regulated cytokines such as TNFα (Tumour necrosis factor α) (*Alsina-Beauchamp et al., 2018*; *Risco et al., 2012*), raising the possibility that the observed effects in p38γ/δKO mice could be due in part to the reduced steady-state levels of TPL2. However, p38γ/p38δ deficiency does not completely recapitulate the effects observed in TPL2-deficient (TPL2KO) mice suggesting specific p38γ/p38δ functions. For example, in a candidiasis mouse model, *Candida albicans* infection and cytokine production is reduced in p38γ/δKO mice, but increased in TPL2KO mice relative to wild-type (WT) (*Alsina-Beauchamp et al., 2018*). Also, in a chemically induced colitis-associated colon cancer model, deletion of p38γ/p38δ decreases tumour development, whereas in TPL2KO mice tumour development is increased (*Koliaraki et al., 2012*; *del Reino et al., 2014*).

To investigate the specific TPL2-independent functions of p38γ/p38δ in the innate immune responses, we generate a *Mapk12*[D171A/D171A]/*Mapk13*[−/−] (p38γ/δKIKO) mouse that expressed a catalytically inactive p38γ isoform and lacks p38δ. Here, we report that, in contrast to p38γ/δKO mice, p38γ/δKIKO mice show normal TPL2 protein levels, and therefore are a good tool to elucidate TPL2-independent roles of p38γ and p38δ in vivo. We studied p38γ/δKIKO mice inflammation and infection in two different mouse models of sepsis, and compared with WT mice. Also, we analysed the gene expression and protein phosphorylation in lipopolysaccharide (LPS)-activated p38γ/δKIKO macrophages, and found that these were different compared to WT macrophages. We demonstrate that p38γ/p38δ phosphorylate the transcription factor myocyte enhancer factor-2D (MEF2D) at Ser444 (Ser437 in mouse), and that this phosphorylation modulates MEF2D's transcriptional activity. MEF2D belongs to MEF2 family, is highly regulated by extracellular stimuli, and has been implicated in transcriptional control of cytokines in innate immune cells (*Lu et al., 2021*; *Wang et al., 2021*). Based on this result and other findings in this study, we demonstrate the importance of p38γ and p38δ for innate immune responses and suggest that pharmacological inhibition of p38γ and p38δ activity would be beneficial in inflammatory diseases and infection.

## Results

### Generation and characterization of *Mapk12*[D171A/D171A], *Mapk13*[−/−] (p38γ/δKIKO) mice

Since p38γ expression in p38γ/δKO cells restores TPL2 levels independently of its kinase activity (*Risco et al., 2012*; *Escós et al., 2022*), we decided to generate a p38γ/δKIKO mouse line by crossing *Mapk12*[D171A/D171A] and *Mapk13*[−/−] mice. p38γ/δKIKO mice, which have catalytically inactive

p38γ and lack p38δ, will not have either p38γ or p38δ kinase activity, as p38γ/δKO mice. Genotype was confirmed by PCR (*Figure 1—figure supplement 1A*). p38γ/δKIKO mice were viable and fertile and had no obvious health problems. Western blot analysis using p38γ and p38δ antibodies confirmed that p38γ/δKIKO mouse embryonic fibroblasts (MEFs) did not express p38δ and expressed p38γ, although to lower levels to that of WT cells (*Figure 1—figure supplement 1B*). The expression levels of p38α, c-Jun N-terminal Kinase (JNK1/2) and extracellular signal-regulated kinase 5 (ERK5) were similar in p38γ/δKIKO and in WT MEFs (*Figure 1—figure supplement 1B*). In response to osmotic stress induced by sorbitol all signalling pathways were activated in p38γ/δKIKO and WT cells (*Figure 1—figure supplement 1B*). Osmotic shock caused p38γ phosphorylation in both WT and p38γ/δKIKO MEF (*Figure 1—figure supplement 1C*); however, phosphorylation of the protein hDlg at Ser158, a physiological p38γ substrate (*Sabio et al., 2005*), was only observed in WT, but not in p38γ/δKIKO MEF (*Figure 1—figure supplement 1D*), confirming that the p38γ expressed in p38γ/δKIKO cells is catalytically inactive.

We next examined Toll-like receptor 4 (TLR4) stimulation by LPS in bone marrow-derived macrophages (BMDMs). Differentiation of bone marrow (BM) progenitor cells to macrophages was not affected in p38γ/δKIKO mice, as indicated by the expression of macrophage-specific membrane protein marker F4/80 (*Figure 2—figure supplement 1A*). We found that WT and p38γ/δKIKO BMDM exhibited similar levels of TPL2, whereas TPL2 expression was severely reduced in p38γ/δKO cells as found previously (*Figure 1A, B*, *Figure 1—figure supplement 1E*). A20 Binding Inhibitor of NF-κB2 (ABIN2), a TPL2-associated protein, was expressed in WT and p38γ/δKIKO macrophages, but was not in p38γ/δKO cells (*Figure 1A*). As TPL2 mediates TLR activation of ERK1/2 signalling in macrophages (*Gantke et al., 2011*), we next analysed the activation of ERK1/2, as well as JNK1/2, p38α and the canonical NF-κB pathway, whose activation is triggered by TLR4 stimulation in macrophages (*Gaestel et al., 2009*; *Lee and Kim, 2007*; *Takeuchi and Akira, 2010*). ERK1/2 activation in p38γ/δKIKO BMDM was similar to that in WT cells (*Figure 1A, B*). We also confirmed that the activation of ERK1/2 was impaired in p38γ/δKO macrophages compared to WT (*Figure 1A, B*). Phosphorylation of NF-κB1 p105 (p105) and activation loop phosphorylation of JNK1/2 and p38α were similar in all three LPS-stimulated WT, p38γ/δKIKO, and p38γ/δKO BMDM (*Figure 1A*). These results indicate that TPL2 protein levels, and therefore activation of the ERK1/2 pathway by TLR4, were regulated by p38γ independently of its kinase activity. Accordingly, we have recently demonstrated that p38γ, and also p38δ, kinase-independent activity regulates the amount of TPL2 protein at two posttranscriptional levels: (1) increasing TPL2 protein stability by binding to the complex TPL2/ABIN2/Nuclear Factor-κB1p105 (NF-κB1p105), and (2) modulating *Map3k8* mRNA translation (*Escós et al., 2022*). The RNA binding protein aconitase-1 (ACO1) interacts with the 3′ untranslated region (UTR) of *Map3k8* mRNA, repressing its translation and decreasing the levels of TPL2 protein in the cells (*Escós et al., 2022*). In the absence of p38δ, p38γ interacts with ACO1 impairing its association with *Map3k8* 3′UTR, which leads to an increase in *Map3k8* mRNA translation (*Escós et al., 2022*).

## p38γ/δKIKO mice are protected from *C. albicans* infection and LPS-induced septic shock

We next used p38γ/δKIKO mice to determine the role of p38γ and p38δ signalling in innate immune responses and inflammation. We performed a comparative analysis of *C. albicans* infection in p38γ/δKIKO and WT mice. In this model, the kidney is the main organ target (*Alsina-Beauchamp et al., 2018*). Fungal burden and total number of infiltrating leucocytes (CD45+ cells) in the kidneys of p38γ/δKIKO mice were lower than in WT control at day 3 post-infection (*Figure 2A, B*). We then checked that resting levels of both macrophages and neutrophils were similar in the BM and spleen of WT and p38γ/δKIKO mice (*Figure 2—figure supplement 1B–D*). Absolute cell numbers in BM and spleen, as well as spleen weight, were similar in both genotypes (*Figure 2—figure supplement 1E, F*). Next, we determined the renal recruitment of macrophages (F4/80+ cells) and neutrophils (Ly6G+ cells), the major leucocyte types involved in *C. albicans*-induced inflammation (*Alsina-Beauchamp et al., 2018*). We found that after *C. albicans* infection, the percentage and the total number of F4/80+ cells were significantly reduced in p38γ/δKIKO mice compared to WT, whereas Ly6G+ cells recruitment was also reduced but not significantly different between genotypes (*Figure 2C*).

The decrease in renal recruitment of leucocytes was paralleled with a reduced chemokine and cytokine expression (*Figure 2D*) as found previously in p38γ/δKO mice (*Alsina-Beauchamp et al., 2018*).

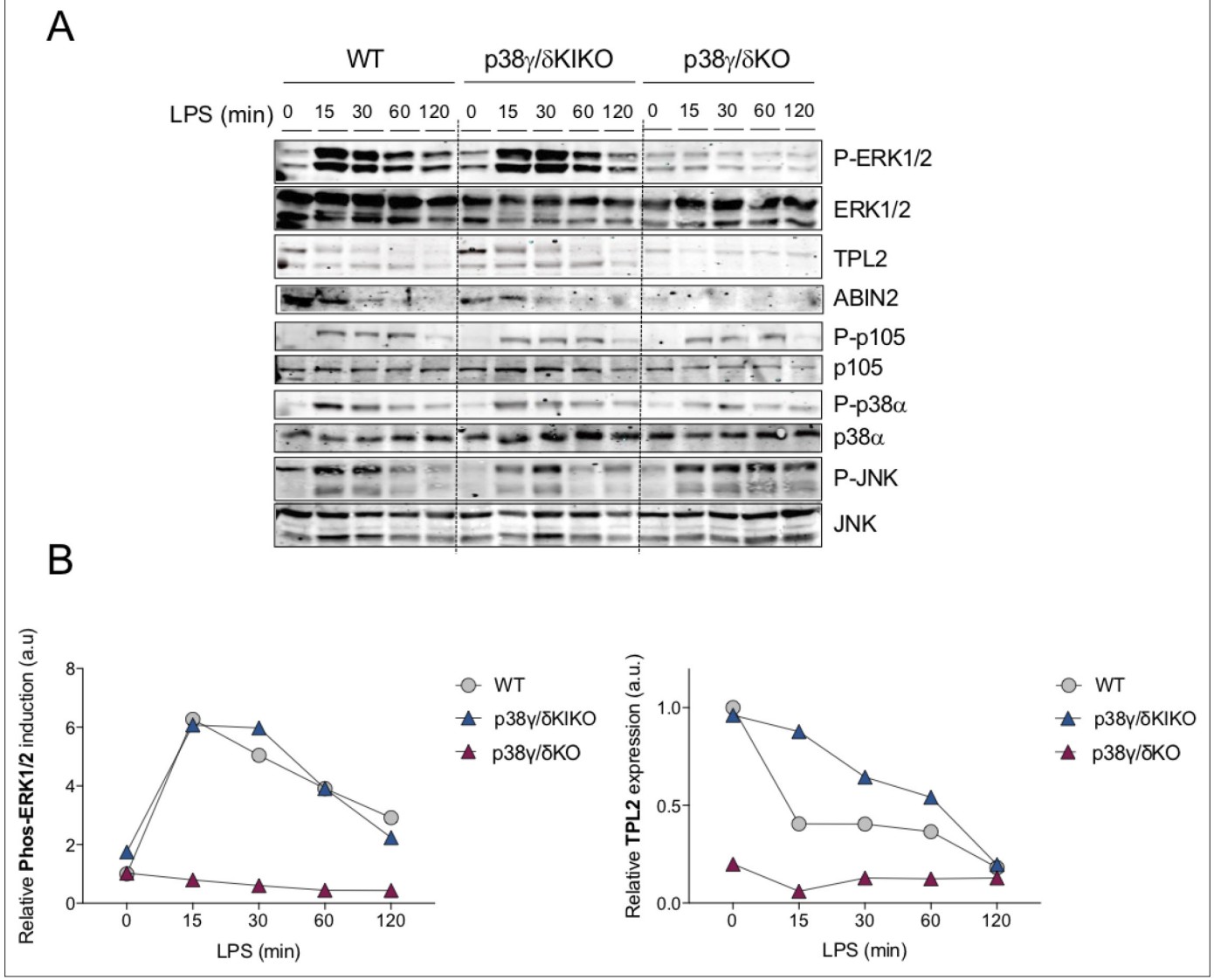

**Figure 1.** Lipopolysaccharide (LPS)-induced ERK1/2 activation in p38γ/δKIKO macrophages. (**A**) Bone marrow-derived macrophage (BMDM) from wild-type (WT), p38γ/δKO, and p38γ/δKIKO mice was exposed to LPS (100 ng/ml) for the indicated times. Cell lysates (30 µg) were immunoblotted with the indicated antibodies to active phosphorylated p38α (P-p38α), JNK1/2 (P-JNK), and ERK1/2 (P-ERK1/2). Phosphorylated p105 (P-p105) and total protein levels of p38α, JNK1/2 (JNK), ERK1/2, TPL2, and p105 were also measured in the same lysates. Representative blots of two independent experiments with similar results are shown. (**B**) Intensity of the bands corresponding to Phos-ERK1/2 and TPL2 (panel A) was quantified. Data relative to WT Time 0 min are shown.

The online version of this article includes the following source data and figure supplement(s) for figure 1:

**Source data 1.** Labelled (.pdf) and raw (.jpeg) western blot images showed in panel A.

**Figure supplement 1.** Characterization of the p38γ/δKIKO mouse.

**Figure supplement 1—source data 1.** Labelled (.pdf) and raw (.tif) western blot images showed in panel A.

**Figure supplement 1—source data 2.** Labelled (.pdf) and raw (.tif,. jpeg) western blot images showed in panel B.

**Figure supplement 1—source data 3.** Labelled (.pdf) and raw (.jpeg) western blot images showed in panel C.

**Figure supplement 1—source data 4.** Labelled (.pdf) and raw (.jpeg) western blot images showed in panel D.

**Figure supplement 1—source data 5.** Labelled (.pdf) and raw (.jpeg) western blot images showed in panel E.

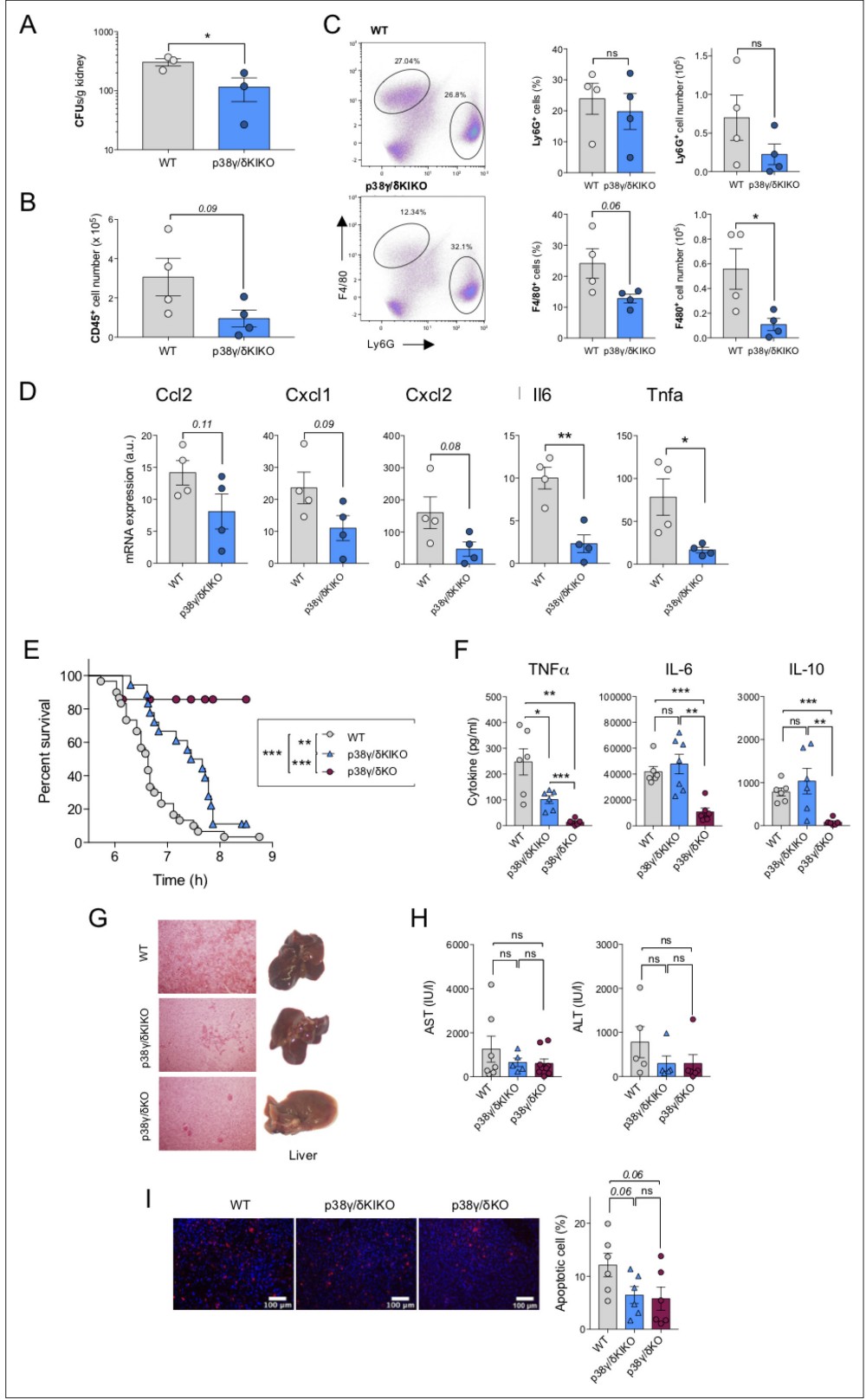

**Figure 2.** Reduced inflammation in p38γ/δ KIKO mice in response to septic shock. (**A**) Wild-type (WT) and p38γ/δKIKO mice were intravenously injected with 1 × 10⁵ CFU of *C. albicans*. Kidney fungal load was determined 3 days after infection. Each symbol represents an individual mouse. Figure shows mean ± standard error of the mean (SEM), ns not significant; *p ≤ 0.05, relative to WT kidney cells. Kidney cells were stained with (**B**) anti-CD45,

*Figure 2 continued on next page*

*Figure 2 continued*

(**C**) anti -Ly6G and -F4/80 antibodies and positive cells analysed by flow cytometry. CD45[+] cells were gated and -F4/80[+] and -Ly6G[+] cells analysed by flow cytometry. Representative profiles are shown. Each symbol represents an individual mouse. Histograms shows mean ± SEM, ns not significant; *p ≤ 0.05, relative to WT kidney cells. (**D**) Mice were treated as in (**A**) and the mRNA levels of indicated genes in the kidney were measured by quantitative PCR (qPCR) 3 days after infection. Each symbol represents an individual mouse. Figure shows mean ± SEM (*n* = 4 mice/condition). ns, not significant, *p ≤ 0.05, **p ≤ 0.01. (**E**) WT (*n* = 31), p38γ/δKO (*n* = 18), and p38γ/δKIKO (*n* = 19) mice were injected with lipopolysaccharide (LPS) (50 µg/kg) and D-Gal (1 g/kg), and survival was monitored for up to 9 hr. Graph shows % survival at the indicated times. **p ≤0.01, ***p ≤ 0.001. (**F**) Serum from mice (**E**) was collected 2 hr after LPS/D-Gal injection, and TNFα, IL-6, and IL10 were measured in a Luminex cytokine assay. Each symbol represents an individual mouse. Figure shows mean ± SEM (*n* = 6–7 mice). ns, not significant; *p ≤ 0.05, **p ≤ 0.01, ***p ≤ 0.001. (**G**) Livers were collected 6 hr after LPS/D-Gal injection. Panels show haematoxylin and eosin (H&E) stained liver sections (left) and whole livers (right). (**H**) Serum ALT (alanine transaminase) and AST (aspartate aminotransferase) activity at 6 hr after LPS/D-Gal injection. Each symbol represents an individual mouse. Figure shows mean ± SEM (*n* = 5–7 mice). ns, not significant. (**I**) Apoptotic TUNEL positive (red) and total nuclei (Hoechst stained-blue) cells were counted using ImageJ programme and the percentage of apoptotic cells calculated. 25 sections per mouse were scored. Representative TUNEL stained liver sections are shown, and figure shows mean ± SEM (*n* = 6 mice). ns, not significant. Each symbol represents an individual mouse.

The online version of this article includes the following figure supplement(s) for figure 2:

**Figure supplement 1.** Characterization of immune cell populations of the p38γ/δKIKO mouse in basal conditions.

Cytokines *Tnfa* and *Il6* mRNA levels were clearly lower in the kidney of p38γ/δKIKO mice than in WT animals at day 3 post-infection (***Figure 2D***). Together, these findings support the role of p38γ/p38δ in the inflammatory response to *C. albicans* by modulating the production of inflammatory molecules and the recruitment of leucocytes into the *C. albicans* infected kidney. These data also show that p38γ/δKIKO behave similar to p38γ/δKO mice in a candidiasis model, as described in ***Alsina-Beauchamp et al., 2018***, and validate p38γ/δKIKO mice as a new tool to study the physio-pathological function of alternative p38MAPK in vivo, independently of TPL2.

Since TPL2 does not play a critical role in *C. albicans* infection (***Alsina-Beauchamp et al., 2018***), we also examined if p38γ/δKIKO and p38γ/δKO mice behave similarly in a TPL2-dependent septic shock model induced by the endotoxin LPS (***Dumitru et al., 2000***). We have previously reported that p38γ/p38δ deficiency protects mice from sepsis induced by LPS (***Risco et al., 2012***). We then treated mice with *Escherichia coli*-derived LPS plus the hepatotoxic compound D-Galactosamine (D-Gal) and checked mouse survival comparing p38γ/δKIKO with WT or p38γ/δKO mice. p38γ/δKIKO mice were significantly more resistant to LPS-induced septic shock than WT mice, although the combined p38γ/p38δ deficiency had a more pronounced protective effect (***Figure 2E***). We have already shown that the reduced susceptibility of p38γ/δKO mice to LPS-induced septic shock was at least in part due to an overall decrease in cytokine production (***Risco et al., 2012***). We observed significantly reduced levels of circulating cytokines IL-6, TNFα, and IL-10 in p38γ/δKO mice than in WT mice after LPS/D-Gal treatment, whereas in p38γ/δKIKO mice only the production of TNFα was significantly reduced compared to WT (***Figure 2F***). Haemorrhage in liver from p38γ/δKIKO mice was lower than WT, but was higher than that observed in p38γ/δKO liver (***Figure 2G***). The circulating serum transaminases, alanine transaminase (ALT), and aspartate aminotransferase (AST), two markers of hepatic necrosis, were noticeably higher in WT than in p38γ/δKO and p38γ/δKIKO mice, although this difference was not statistically significant due to mouse individual variation (***Figure 2H***). In addition, TUNEL analysis showed a decreased apoptosis in p38γ/δKO and p38γ/δKIKO mice compared to WT (***Figure 2I***), indicating a reduction in liver cell death and protection against liver damage in the mutant mice.

These comparative experiments in p38γ/δKIKO and p38γ/δKO mice show that the combined deletion of p38γ/p38δ has a more pronounced impact in LPS-induced cytokine production in mice. This indicates that the effects observed in p38γ/δKO mice are partly due to the decreased TPL2 levels and the blockade of ERK1/2 pathway signalling, and also confirms a role for p38γ/p38δ, independently of TPL2 expression, in regulating LPS/D-Gal-induced septic shock by increasing liver damage and TNFα production, and in *C. albicans* infection modulating the inflammatory response.

## Analysis of TLR4-induced gene expression in p38γ/δKIKO macrophages

Since p38γ/p38δ are important in the immune response to septic shock and to the TLR4 ligand LPS in the macrophage (*Alsina-Beauchamp et al., 2018*; *Risco et al., 2012*), we stimulated WT and p38γ/δKIKO BMDM with LPS and analysed gene expression by RNA-sequencing (RNA-Seq) to identify target genes specifically affected by the lack of p38γ kinase activity and p38δ expression. Both, WT and p38γ/δKIKO BMDM, showed a strong transcriptional response at 30 and 60 min of stimulation with LPS (*Figure 3A–C*). Different gene profiles of p38γ/δKIKO and WT BMDM at basal conditions (time 0) indicate that p38γ and p38δ also regulate gene expression in resting conditions (*Figure 3B*). WT showed increased (at least 2.0-fold, p-value <0.05) expression of 197 and 533 genes at 30 and 60 min, respectively, whereas p38γ/δKIKO BMDM showed increased expression of 264 and 647 genes at 30 and 60 min, respectively (*Figure 3A*). Venn diagram analysis revealed that ~60% of the upregulated genes were common between WT and p38γ/δKIKO BMDM at both 30 and 60 min; whereas ~30–40% of the down-regulated genes were shared between genotypes (*Figure 3C*).

We next compared gene expression profiles between genotypes in LPS-stimulated p38γ/δKIKO and WT BMDM (*Figure 3D*) and selected genes with either increased or decreased expression between genotypes by at least 1.5-fold (−1.5 > logFC > 1.5), with a p-value <0.05 (*Figure 3D*). Gene Ontology (GO) analyses of differentially expressed genes using DAVID (the Database of Annotation, Visualization and Integrated Discovery) revealed the enrichment for genes implicated in the innate immune and inflammatory response (*Figure 3E*, *Figure 3—figure supplement 1A*).

One of the main processes affected in LPS-stimulated p38γ/δKIKO was the immune response and cytokine production. Therefore, to validate the RNA-Seq analysis we performed a comparative analysis by quantitative PCR (qPCR) of the expression of genes involved in these processes (*Figure 3F*, *Figure 3—figure supplement 1A*). We found that after LPS treatment the expression of *Ccl5*, *Cxcl9* and *Nos2* mRNAs increased in p38γ/δKIKO macrophages compared to WT, as observed in RNA-Seq analysis. *Il1b* and *Cxcl10* mRNA expression was also significantly higher at 30 and 60 min after LPS treatment in p38γ/δKIKO BMDM. Whereas, *Ifnb1* and *Il12a* mRNA expression, which are TPL2-ERK1/2 targets, and *Il6* and *Tnfa* mRNA expression were similar in p38γ/δKIKO and WT macrophages (*Figure 3F*). These results indicate that p38γ and p38δ in macrophages specifically regulate production of inflammatory molecules in response to LPS.

We also assessed cytokines expression levels using a Mouse Cytokine Array, at 6 hr after LPS treatment. We found that the global cytokine expression pattern in WT and p38γ/δKIKO macrophages seemed similar (*Figure 3—figure supplement 1B*); nonetheless, after quantification we found that the levels of CXCL1, CCL2, CCL12, CXCL2, CCL5, IL-5, IL-10, sICAM-1, IL-12p70, and IL-1α were significantly higher in p38γ/δKIKO cells (*Figure 3—figure supplement 1C*). We confirmed that IL-1β production was increased in p38γ/δKIKO macrophages compared to WT, whereas TIMP-1, TREM-1, and IL-6 were decreased (*Figure 3—figure supplement 1C*). These data indicate that p38γ and p38δ control the production of inflammatory molecules in response to LPS in macrophages by modulating cytokine transcription and synthesis.

## Identification of p38γ- and p38δ-dependent protein phosphorylation

To establish the molecular mechanism by which p38γ/p38δ modulate macrophage activation and cytokine production, we sought to identify the p38γ/p38δ substrates by performing a comparative phosphoproteomic analysis of LPS-stimulated WT, p38γ/δKIKO, and p38γ/δKO macrophages. We used peritoneal macrophages since p38γ and p38δ are expressed at much higher levels in these macrophages than in BMDM (*Risco et al., 2012*). We first confirmed ERK1/2 and p38α phosphorylation in all LPS-stimulated macrophages (*Figure 4—figure supplement 1A*). p38α activation was similar in all genotypes, whereas ERK1/2 phosphorylation was impaired in p38γ/δKO macrophages and slightly reduced in p38γ/δKIKO cells compared to WT (*Figure 4—figure supplement 1A*). Additionally, the LPS-induced cytokine production pattern in peritoneal macrophages was comparable to that of BMDM (*Figure 4—figure supplement 1B*).

Comparison of the phosphoproteomes of LPS-stimulated WT, p38γ/δKO, and p38γ/δKIKO with control unstimulated macrophages confirmed the phosphorylation of proteins from the classical/canonical TLR4-activated signalling pathways such as p38α, its substrate the kinase MAPKAPK2 (MK2) and the protein Tristetraproline (TTP), which is itself a MK2 substrate (*Figure 4—figure supplement 1C*). This indicates the robustness of our experimental approach. Heatmap analysis revealed that

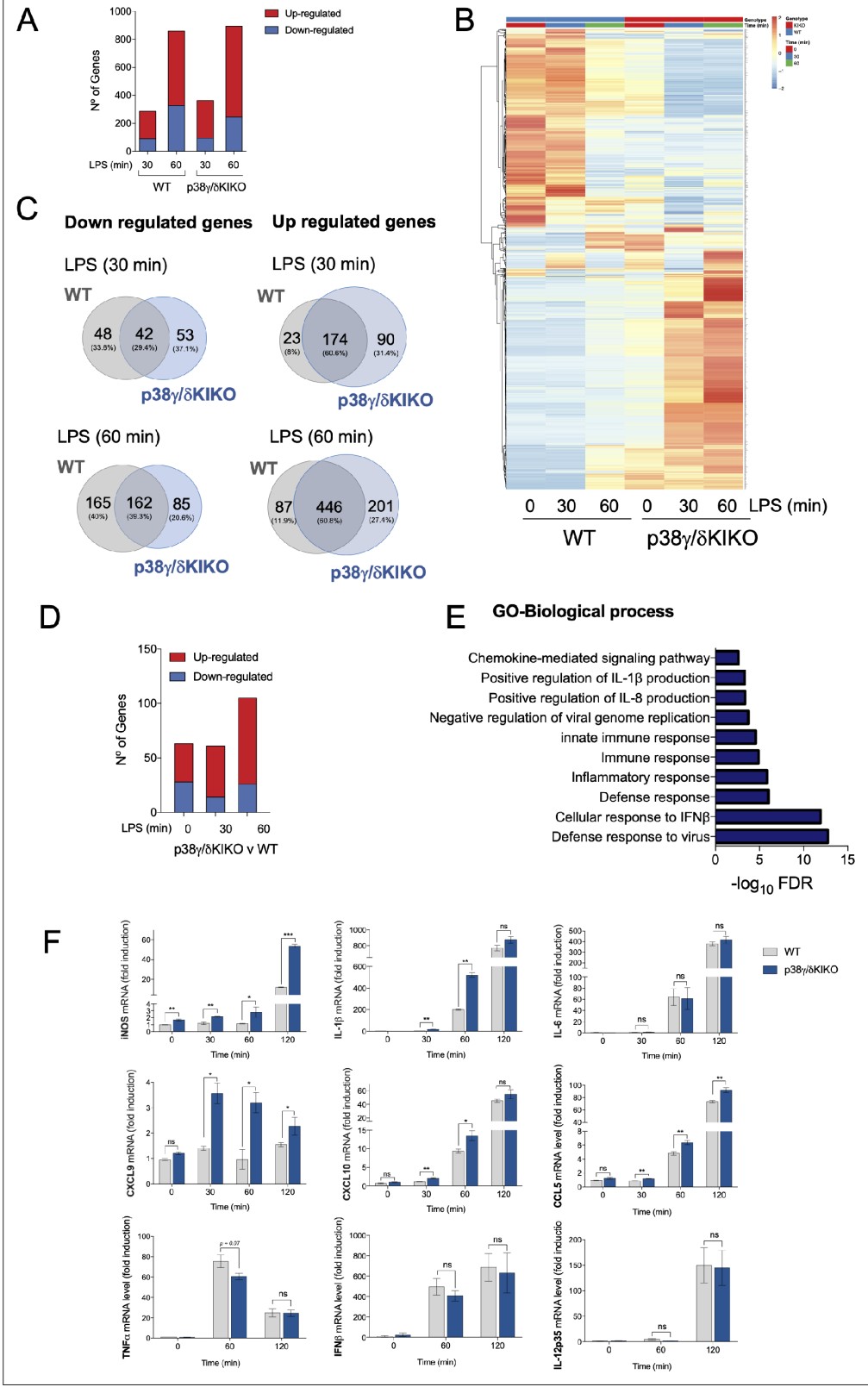

**Figure 3.** RNA-sequencing analysis in lipopolysaccharide (LPS)-stimulated wild-type (WT) and p38γ/δKIKO macrophages. (**A**) Bone marrow-derived macrophage (BMDM) from WT and p38γ/δKIKO mice was exposed to LPS (100 ng/ml) for 0, 30, and 60 min and gene expression analysed by RNA-sequencing. Bar plot showing the number of differentially expressed genes up- or down-regulated after LPS stimulation ($-2 > \log FC > 2$, p-value <0.05).

*Figure 3 continued on next page*

*Figure 3 continued*

Three samples per condition were used. (**B**) Hierarchical heatmap of the differentially expressed genes in panel (**A**). (**C**) Venn diagrams showing the overlaps of genes up- or down-regulated over the time course of LPS stimulation in p38γ/δKIKO and WT macrophages. (**D**) Bar plot showing the number of differentially expressed genes up- or down-regulated (−1.5 > logFC > 1.5, p-value <0.05) in p38γ/δKIKO macrophages compared with WT after LPS stimulation at the indicated times (minutes). Three samples per condition were used. (**E**) Enrichment analysis of Gene Ontology (GO) biological processes of the differentially expressed genes in LPS-activated p38γ/δKIKO macrophages after LPS stimulation, excluding genes that are common with LPS-treated WT macrophages. (**F**) BMDM from WT and p38γ/δKIKO mice was exposed to LPS (100 ng/ml) for the indicated times. mRNA expression of indicated genes at different times, relative to WT expression at 0 hr, was determined by quantitative PCR (qPCR) and normalized to *Actb* mRNA. Data show mean ± standard error of the mean (SEM) (*n* = 3–6). ns, not significant; *p ≤ 0.05; **p ≤ 0.01, ***p ≤ 0.001, relative to WT mice in the same conditions.

The online version of this article includes the following source data and figure supplement(s) for figure 3:

**Source data 1.** Excel files of data (raw and processed) showed in panels A–E.

**Figure supplement 1.** RNA-sequencing analysis and cytokine production in lipopolysaccharide (LPS)-stimulated wild-type (WT) and p38γ/δKIKO macrophages.

**Figure supplement 1—source data 1.** Excel files of data showed in panel A.

**Figure supplement 1—source data 2.** Labelled (.pdf) and raw (.tif) images from the arrays showed in panel B.

peptide phosphorylation patterns were similar in p38γ/δKO and p38γ/δKIKO LPS-stimulated macrophages, as compared to WT (*Figure 4A*). Comparison with the WT showed significant reduction of 35 and 16 phosphopeptides in p38γ/δKIKO and p38γ/δKO LPS-treated samples, respectively (−1.0 > logFC > 1.0, with a p-value <0.05) (*Figure 4B–D*). Notably, we observed that 158 and 94 phosphopeptides were increased in p38γ/δKIKO and p38γ/δKO samples, respectively (−1.0 > logFC > 1.0, with a p-value <0.05) (*Figure 4B–D*). The number of up-regulated phosphopeptides was larger than those down-regulated indicating that, upon LPS stimulation, p38γ and p38δ control the activity of a yet unidentified kinase, or phosphatase, or both, in macrophages. Venn diagram analysis revealed that only the phosphorylation of peptides from nucleolar protein 56 (Nop56), Myocyte enhancer factor 2D (MEF2D) and osteopontin (spp1), was down-regulated in both p38γ/δKIKO and p38γ/δKO macrophages (*Figure 4D*).

We next examined the down-regulated phosphorylation sites lying within the well-characterized p38MAPK phosphorylation site consensus S/T-P motif (*Cuenda and Rousseau, 2007*), as these represent likely candidates for direct p38γ and p38δ physiological substrates. We identified 4 candidates with S/T-P phosphorylation site in p38γ/δKO v WT comparison, and 11 in p38γ/δKIKO v WT comparison, one of them being the previously described p38δ substrate, Stathmin (Stmn1) (*Cuenda and Rousseau, 2007*; *Figure 4—figure supplement 2*). MEF2D (Ser437 in mouse and Ser444 in human) was the only protein identified in both p38γ/δKO v WT and p38γ/δKIKO v WT comparisons. MEF2D is a member of the transcription factor MEF2 family, which contains MEF2A, B, C, and D, that were initially identified in muscle, but are also expressed in neurons, and cells of the immune system such as T and B cells and myeloid cells (*McKinsey et al., 2002*).

The activity of the MEF2 transcription factors is regulated by p38α-mediated phosphorylation (*McKinsey et al., 2002*); however, to our knowledge, the phosphorylation of MEF2D by p38γ and/or p38δ in cells has not been demonstrated. We then checked if MEF2D was phosphorylated by p38γ and/or p38δ using γ$^{32}$P-ATP in in vitro kinase assay, and compared with the phosphorylation by p38α and p38β. We found that MEF2D was equally phosphorylated by p38α and p38δ, and less by p38γ, whereas p38β weakly phosphorylated MEF2D (*Figure 4E, F*). The protein control myelin basic protein (MBP) was equally phosphorylated by all p38s (*Figure 4G*). We determined the MEF2D sites phosphorylated by p38α and p38δ, in vitro, by mass spectrometry analysis. We identified ten amino acids phosphorylated by these p38MAPKs, although the p38δ phosphorylation sites within MEF2D differed from those phosphorylated by p38α (*Table 1*). Four of the sites were phosphorylated by the two kinases, whereas Ser98 and Ser192 were exclusively phosphorylated by p38α, and Ser121, Ser275, Ser444, and Ser472 were specific for p38δ (*Table 1*, *Figure 4H*). This confirmed that Ser444 (Ser 437 in mouse) was one of MEF2D phosphorylation sites specific for p38δ.

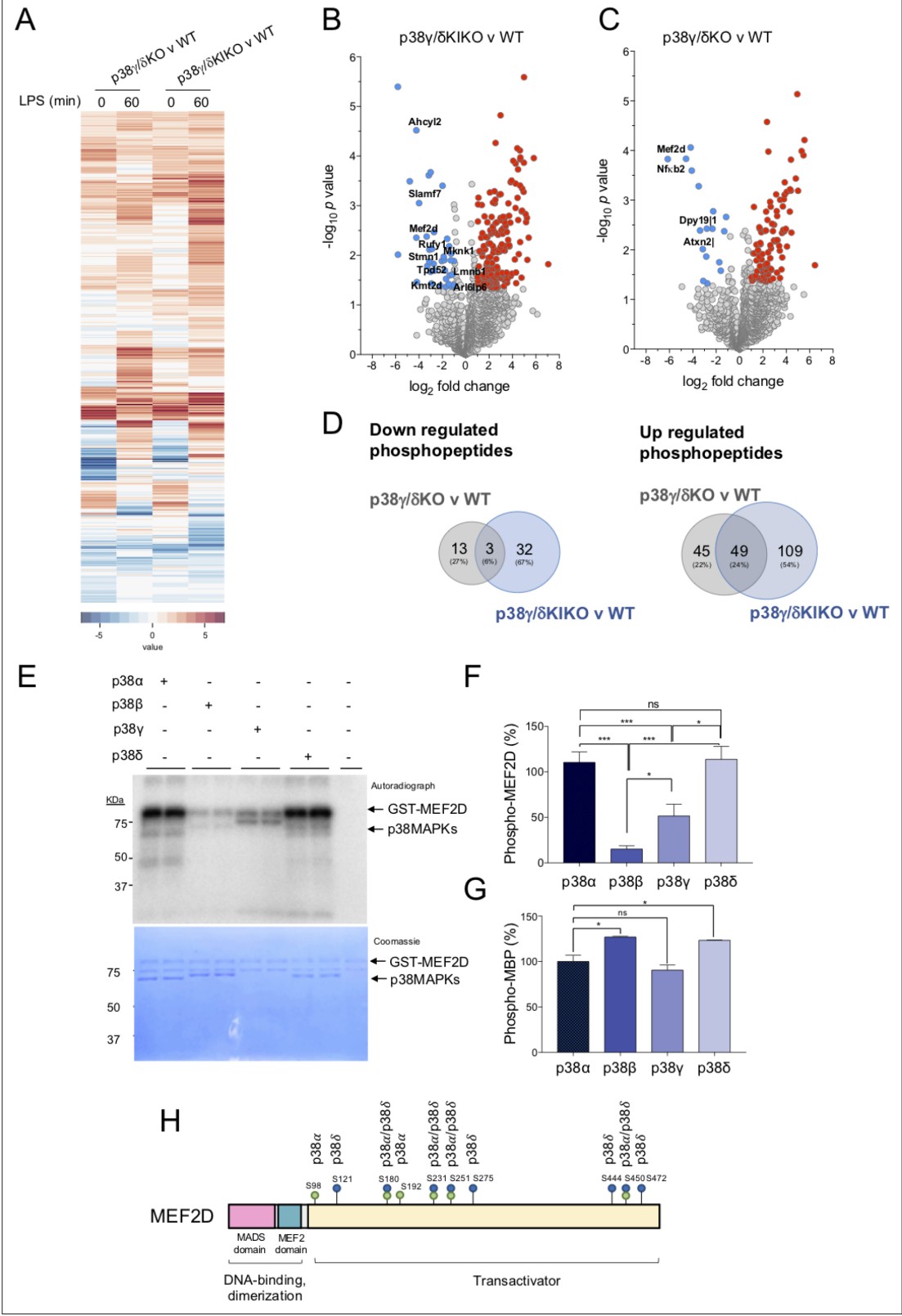

**Figure 4.** Identification of proteins phosphorylated by p38γ/p38δ. (**A**) Peritoneal macrophages from wild-type (WT), p38γ/δKO, and p38γ/δKIKO mice were exposed to lipopolysaccharide (LPS) (100 ng/ml) for 0 or 60 min and phosphorylated peptides identified. Heatmap showing differentially phosphorylated phosphosites up- or down-regulated after LPS stimulation. Volcano plots showing the differential phosphoproteome and statistical significance between (**B**) p38γ/δKIKO and WT ($n$ = 3), and (**C**) p38γ/δKO and WT ($n$ = 3) macrophages treated with LPS for 60 min. Examples of

*Figure 4 continued on next page*

*Figure 4 continued*

significantly enriched up-regulated (red) and down-regulated (blue) phosphosites are shown (−1.0 > logFC > 1.0, p-value <0.05). (**D**) Venn diagrams showing the overlaps of differentially phosphorylated proteins (up- or down-regulated) at 60-min LPS stimulation in p38γ/δKIKO and p38γ/δKO macrophages compared to WT. (**E**) Recombinant GST-MEF2D was incubated with active recombinant p38α, p38β, p38γ, or p38δ for 60 min in a phosphorylation reaction mix containing Mg-[γ³²P]ATP, as described in Materials and methods. The activity of recombinant p38α, p38β, p38γ, or p38δ was matched using myelin basic protein (MBP) as substrate and 0.5 U/ml were used in the assay. Reaction was stopped with sodium dodecyl sulfate (SDS)-sample buffer. Samples were resolved in SDS–polyacrylamide gel electrophoresis (PAGE), subjected to Coomassie blue staining and autoradiography. (**F**) Bands corresponding to ³²P-MEF2D from panel (**E**) were quantified. Data show mean ± standard error of the mean (SEM) from two experiments in duplicate. ns, not significant *p ≤ 0.05; ***p ≤ 0.001, relative to MEF2D phosphorylation by p38α. (**G**) Recombinant MBP was incubated with active (0.5 U/ml) recombinant p38α, p38β, p38γ, or p38δ for 60 min in a phosphorylation reaction mix containing Mg-[γ³²P]ATP, as described in Materials and methods. (**H**) Schematic representation of the sites in MEF2D phosphorylated by p38α and/or p38δ, all of them located in the transactivation domain.

The online version of this article includes the following source data and figure supplement(s) for figure 4:

**Source data 1.** Excel file of data showed in panels A–D and in *Figure 4—figure supplement 1* panel C.

**Figure supplement 1.** Identification of proteins phosphorylated by p38γ/p38δ.

**Figure supplement 1—source data 1.** Labelled (.pdf) and raw (.jpeg) western blot images showed in panel A.

**Figure supplement 2.** Identification of myocyte enhancer factor-2D (MEF2D) residues phosphorylated in vitro by p38α and p38δ.

**Figure supplement 2—source data 1.** Excel file of data showed in *Figure 4—figure supplement 2*.

**Table 1.** Sites on myocyte enhancer factor-2D (MEF2D) phosphorylated by p38α or p38 δ in vitro. Human GST-MEF2D was incubated in a phosphorylation reaction mix containing Mg-ATP in the absence (control) or the presence of p38α or p38 δ, and subjected to sodium dodecyl sulfate–polyacrylamide gel electrophoresis (SDS–PAGE). Phosphorylated MEF2D was excised from the gel, digested with trypsin and peptides separated as indicated in Materials and methods. Phosphorylated residues were identified by liquid chromatography–mass spectrometry (LC–MS)/MS. Peptides containing the phosphorylated residues are shown. Phosphorylated residues are indicated as pS in red, deaminated residues are indicated as (deam) in blue. (-) no or poor phosphorylation detected; (+) phosphorylation detected; (++) strong phosphorylation detected.

| Protein phospho-site (Q14814) | Phosphopeptide | Control | p38α | p38δ |
|---|---|---|---|---|
| | KGFN(deam)GCD**pS**PEPDGEDSLEQSPLLEDK | + | +++ | - |
| | KGFNGCD**pS**PEPDGEDSLEQSPLLEDK | | | |
| | GFNGCD**pS**PEPDGEDSLEQSPLLEDK | | | |
| S98 | GFNGCD**pS**PEPDGEDSLEQSPLLEDKYR | | | |
| S121 | RA**pS**EELDGLFR | + | + | +++ |
| S180 | LL**pS**PQQPALQR | - | +++ | +++ |
| S192 | NSV**pS**PGLPQR | - | + | - |
| S231 | A**pS**PGLLPVANGNSLNK | - | + | +++ |
| | A**pS**PGLLPVAN(deam)GNSLNK | | | |
| | A**pS**PGLLPVAN(deam)GN(deam)SLNK | | | |
| S251 | VIPAK**pS**PPPPTHSTQLGAPSR | | | |
| | **pS**PPPPTHSTQLGAPSR | - | +++ | +++ |
| S275 | VIT**pS**QAGK | - | - | +++ |
| S444 | SEPV**pS**PSRER | - | - | +++ |
| S450 | ER**pS**PAPPPPAVFPAAR | - | + | +++ |
| S472 | PEPGDGLS**pS**PAGGSYETGDR | - | - | +++ |

## p38γ and p38δ regulate MEF2D transcriptional activity

MEF2D is activated by LPS, and regulates inflammation by modulating the expression of proinflammatory factors. Thus, MEF2D decreases *Nlrp3*, *Nos2*, and *Il1b* expression in microglia, and *Il10* transcription in macrophages (*Lu et al., 2021*; *Yang et al., 2015*; *Pattison et al., 2020*). Transcriptional activity of MEF2D can be partially modulated by post-translational modifications such as phosphorylation. However, the kinases that regulate MEF2D phosphorylation and function are still largely unknown. To investigate whether p38γ and p38δ modulate MEF2D transcriptional activity, we examined the TLR4-induced expression of *Jun*, *Cd14*, and *Hdac7* genes, whose transcription is regulated by MEF2D (*Lu et al., 2021*; *Han and Prywes, 1995*; *Park et al., 2002*). qPCR analysis showed that LPS-induced *Jun*, *Cd14*, and *Hdac7* mRNAs were significantly increased in p38γ/δKIKO macrophages compared with WT cells (*Figure 5A*).

To further examine the possible role of p38γ/p38δ in the specific transcriptional activity of MEF2D we performed reporter assays using a plasmid encoding a *Firefly Luciferase* gene under the control of MEF2 response elements. p38γ/δKIKO, p38γ/δKO, and WT fibroblasts were transfected either with the reporter plasmid alone, or with the reporter plasmid and a plasmid encoding Flag-MEF2D. We observed that the relative luciferase activity was significantly higher when cells, regardless of the genotype, were transfected with MEF2D. In both conditions, we found a marked increase in luciferase activity in p38γ/δKO and p38γ/δKIKO cells compared to WT cells (*Figure 5B*). MEF2D expression levels were similar under each experimental condition (*Figure 5B*). Interestingly, we observed that the transcriptional activity of MEF2D was significantly much higher in p38γ/δKO than in p38γ/δKIKO cells suggesting that, in addition to phosphorylation, the interaction of p38γ with MEF2D or with other protein might be affecting MEF2D transcriptional activity; this question could be addressed in future works. Next, we analysed the effect of the p38MAPK inhibitor BIRB796 on MEF2D transcriptional activity. Since this compound inhibits all four (p38α, p38β, p38γ, and p38δ) p38MAPK isoforms, we also used in parallel the compound SB203580 that blocks p38α and p38β only (*Kuma et al., 2005*), so we could determine which effect was caused by p38α/p38β or by p38γ/p38δ inhibition. Incubation with BIRB796, but not with SB203580 significantly increased luciferase activity in WT cells (*Figure 5C*), indicating that p38γ and p38δ activity regulates MEF2D-mediated transcription. In p38γ/δKO cells, luciferase activity was significantly and equally increased by SB20358 and BIRB796, suggesting that p38α can control MEF2D activity in the absence of p38γ/p38δ. The expression levels of transfected MEF2D were similar under each experimental condition (*Figure 5C*).

MEF2D transcriptional activity is modulated by phosphorylation, and phosphorylation at different MEF2D sites has different effects in its transcriptional activity (*Tang et al., 2005*; *Ke et al., 2015*). For example, MEF2D transcriptional activity is increased after phosphorylation at Thr259, Ser275, Ser294, and Ser314 by the kinase Ataxia telangiectasia mutated (ATM) in neurons (*Chan et al., 2014*), or at Ser179 by ERK5 in Hela cells (*Kato et al., 2000*). However, phosphorylation of MEF2D at Ser251 by DYRK1, in HEK293 cells (*Wang et al., 2021*), or of Ser444 by Cdk5, has been implicated in the repression of MEF2D transcriptional activity in neurons (*Tang et al., 2005*). Particularly, it has been shown that Ser444 phosphorylation by Cdk5 is required for the sumoylation of Lys439, which regulates MEF2D transcriptional activity (*Grégoire et al., 2006*). MEF2D-Ser444 was phosphorylated in vitro by p38δ, and phosphorylation at this residue was decreased in p38γ/δKO and p38γ/δKIKO cells compared to WT cell (*Figure 4B, C*). Consistently, the expression of *Jun*, *Cd14*, *Hdac7*, *Nos2*, and *Il1b* mRNA, which are MEF2D-regulated genes (*Lu et al., 2021*; *Han and Prywes, 1995*; *Park et al., 2002*), is increased by the lack of p38γ/p38δ activity. We then mutated Ser444 to Ala to generate MEF2D-Ser444Ala (MEF2D[S444A]) mutant and study its effect on MEF2D transcriptional activity in WT cells. Transfection of MEF2D[S444A] mutant caused a significant increase in luciferase activity compared to MEF2D (*Figure 5D*), similarly to that observed in p38γ/δKIKO cells (*Figure 5B*). The expression levels of transfected MEF2D and MEF2D-S444A were similar (*Figure 5D*). In addition, qPCR analysis showed that *Jun*, *Cd14*, and *Hdac7* mRNAs as well as *Nos2* and *Il1b* mRNA levels were significantly increased in WT cells expressing MEF2D-S444A compared to WT cells expressing MEF2D (*Figure 5E*). Altogether, these data indicate that phosphorylation at Ser444 represses MEF2D transcriptional activity.

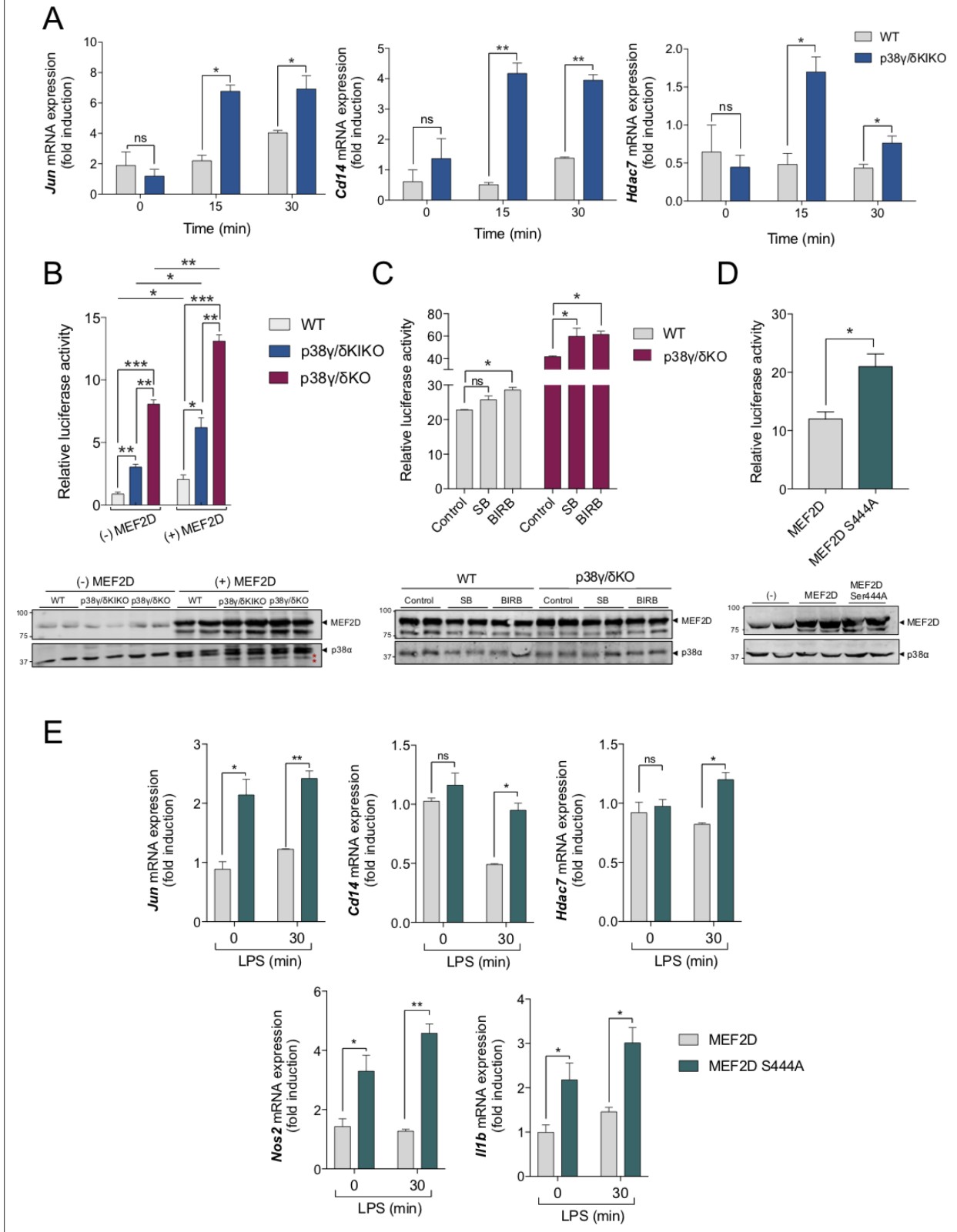

**Figure 5.** Regulation of myocyte enhancer factor-2D (MEF2D) transcriptional activity by p38γ and p38δ and S444 phosphorylation. (**A**) Bone marrow-derived macrophage (BMDM) from wild-type (WT) and p38γ/δKIKO mice was exposed to lipopolysaccharide (LPS) (100 ng/ml) for the indicated times. Relative mRNA expression of *Jun*, *Cd14*, and *Hdac7* genes at different times was determined by quantitative PCR (qPCR). Results were normalized to *Actb* RNA expression and fold induction was calculated relative to WT expression at 0 min. Figure shows mean ± standard error of the mean (SEM)

*Figure 5 continued on next page*

*Figure 5 continued*

from two experiments in triplicate. ns, not significant; *p ≤ 0.05 and **p ≤ 0.01, relative to WT mice in the same conditions. (**B**) WT, p38γ/δKIKO, and p38γ/δKO fibroblasts were co-transfected either with plasmids coding for Flag-MEF2D, Renilla, and luciferase firefly gene under the control of MEF2 response elements ((+) MEF2D), or with plasmids encoding for Renilla and luciferase firefly ((−) MEF2D). (Upper panel) Luciferase activity values were normalized against Renilla. Figure shows mean ± SEM from one representative experiment in triplicate. This experiment was repeated twice more with similar result. ns, not significant; *p ≤ 0.05; **p ≤ 0.01; ***p ≤ 0.001. (Lower panel) Cell lysates (10 µg) were immunoblotted with the indicated antibodies to total protein levels of MEF2D and p38α. Representative blots of two independent experiments with similar results are shown. Molecular weights are indicated. (*) unspecific band (**C**) WT and p38γ/δKO fibroblasts were transfected as in (**B**). Cells were incubated with the indicated p38MAPK inhibitor (or Dimethyl sulfoxide [DMSO] as control) for 6 hr before lysis. SB203580 (SB) at 10 µM and BIRB796 (BIRB) at 10 µM. (Upper panel) Luciferase activity was calculated as in (**B**). (Lower panel) Cell lysates (10 µg) were immunoblotted as in panel B. (**D**) WT fibroblasts were transfected with plasmid Flag-MEF2D or Flag-MEF2D$^{S444A}$, plus plasmids coding Renilla and luciferase firefly gene under the control of MEF2 response elements. (Lower panel) Luciferase activity values were normalized against Renilla. Figure shows mean ± SEM from one representative experiment in triplicate. This experiment was repeated twice more with similar result. *p ≤ 0.05. (Lower panel) Cell lysates (50 µg) were immunoblotted as in panel B. (**E**) WT fibroblasts were transfected with Flag-MEF2D or Flag-MEF2D$^{S444A}$, and stimulated with LPS for 30 min. Relative mRNA expression of the indicated genes was determined by qPCR. Results were normalized to *Actb* RNA expression and fold induction was calculated relative to that in cells expressing Flag-MEF2D WT at time 0 of LPS. Figure shows mean ± SEM from two experiments in triplicate. *p ≤ 0.05.

The online version of this article includes the following source data for figure 5:

**Source data 1.** Labelled (.ppt) western blot images showed in panel B.

**Source data 2.** Labelled (.ppt) western blot images showed in panel C.

**Source data 3.** Labelled (.ppt) western blot images showed in panel D.

# Discussion

Our results determine that p38γ/δKIKO mice could be an excellent tool to elucidate the in vivo roles of p38γ/p38δ, as their use helped us to establish specific roles for p38γ/p38δ in immune and inflammatory responses independent of TPL2. Mice and cells expressing inactive kinases are powerful tools: (1) in the study of the role of kinase binding interactions that are independent of the kinase catalytic activity, (2) to also avoid unspecific side effects caused by the treatment with small molecule inhibitors, and (3) to overcome the problems of protein compensation, due to kinase functional redundancy, inherent in assessing kinase function using knock-out mice or cells.

p38MAPK are a clear example of highly related family members with functional redundancies, which may account for the difficulty on finding altered phenotypes in the p38γ- or p38δ-null mice. This highlights the need for using mice expressing inactive kinases in combination with knock-out mice in the study of the role of these kinases. Here we demonstrate with the use of p38γ/δKIKO mice, that p38γ/p38δ are implicated in the immune response modulating the expression of inflammatory genes and the phosphorylation of regulatory proteins such as the transcription factor MEF2D. Phosphorylation analysis revealed that p38δ phosphorylated recombinant MEF2D more efficiently than p38γ, suggesting that MEF2D is a better substrate for p38δ than for p38γ, at least in vitro. These data support that despite the functional redundancies of p38δ and p38γ, there are also specific substrates, and therefore biological roles, for each kinase. For example, we have recently shown that p38δ is a better kinase phosphorylating the mitogen- and stress-activated kinase (MSK) 1 than p38γ, in vitro. Also, p38δ mediates the phosphorylation and activation of MSK1, in LPS-stimulated macrophages, as well as the phosphorylation of the MSK1 substrates, the transcription factors CREB and ATF1, and the expression of MSK1-dependent genes such as *Dusp1* and *Il1ra* (*Diaz-Mora et al., 2023*). All these evidence suggest that p38δ specifically regulates the production of some inflammatory proteins via MSK1 and also MEF2D, whereas p38γ may have different functions through the phosphorylation of other substrates.

Research using p38γ/δKIKO mice, or alternatively a mouse expressing both inactive p38γ and p38δ, will be of great help to investigate which inflammatory diseases are dependent on p38γ/p38δ catalytic activity. This aspect is very important for the development of p38γ/p38δ inhibitors to treat inflammatory diseases and also cancer.

**Table 2.** Information about the antibodies used in this study.

| Antibody | Name | Provider | Method | Dilution |
|---|---|---|---|---|
| P-ERK1/2 | Phospho-p44/42 MAPK (Erk1/2) (Thr202/Tyr204) Antibody | Cell Signaling | WB | 1/2000 |
| ERK1/2 | p44/42 MAPK (Erk1/2) Antibody | Cell Signaling | WB | 1/1000 |
| TPL2 | Cot (M-20) | Santa Cruz Biotechnology, Inc | WB | 1/1000 |
| ABIN2 | ABIN2 antibody | S.C. Ley laboratory | WB | 2 µg/ml |
| P-p38 | Phospho-p38 MAP Kinase (Thr180/Tyr182) Antibody | Cell Signaling | WB | 1/1000 |
| p38α | p38α (C-20) | Santa Cruz Biotechnology, Inc | WB | 1/1000 |
| p38γ | p38γ (GST-SAPK3) S524A 1st Bleed | DSTT* | WB | 1/1000 |
| p38δ | p38δ (GST-SAPK4) S526A 3rd Bleed | DSTT* | WB | 1/1000 |
| P-JNK1/2 | Rabbit (polyclonal) Anti-JNK1&2 [pTpY183/185] Phosphospecific Antibody, Unconjugated | Biosource | WB | 1/1000 |
| JNK1/2 | SAPK/JNK | Cell Signaling | WB | 1/1000 |
| P-p105 | Phospho-NF-$\kappa$B p105 (Ser933) (18E6) Rabbit mAb | Cell Signaling | WB | 1/1000 |
| p105 | NF-$\kappa$B1 p105 Antibody | Cell Signaling | WB | 1/1000 |
| hDlg | hDlg (SAP97) antibody | DSTT* | WB | 1/1000 |
| P-hDlg (S158) | Phospho-hDlg (SAP97) Ser$^{158}$ | DSTT* | WB | 1/1000 |
| hDlg | hDlg (SAP97) antibody | DSTT* | IP | 1 µg/IP |
| p38γ | p38γ (SAPK3 C-Terminal) [KPPRNLGARVPKETA] | DSTT* | IP | 2 µg/IP |
| CD45 | CD45 Monoclonal Antibody (30-F11), APC #17-0451-82 | eBioscience | FAC | 1/100 |
| F4/80 | F4/80 Monoclonal Antibody (BM8) FITC #11-4801-85 | eBioscience | FAC | 1/100 |
| Ly6G | PE Rat Anti-Mouse Ly-6G Clone 1A8 (RUO) #551461 | BD Bioscience | FAC | 1/50 |

*DSTT (Dundee): Division of Signal Transduction Therapy; University of Dundee (Dundee, UK).

## Materials and methods

### Generation and genotyping of *MAPK12*[D171A/D171A]/*MAPK13*[−/−] (p38γ/δKIKO) mice

p38γ/δKIKO mouse were produced by crossing *MAPK12*[D171A/D171A] and *MAPK13*[−/−] mice and genotype was confirmed by PCR as described elsewhere (*Sabio et al., 2005*; *Sabio et al., 2010*). Briefly, genomic DNA purified from tail biopsy sample was used as a template for PCR, resolved on a 2% agarose gel and examined by greenSafe Premium staining. PCR primers used for *MAPK12*[D171A/D171A] mice genotyping were:

> 5′CAGACCAGACTGGCCTTGAATCCATAGAGATC
> and 5′CTCTGCAGGCACCGAGTACACAGGTGGTGT.
> PCR primers used for *MAPK13*[−/−] mice were:
> 5′CCCTTGAGCCATAGATCCTGGACTTTGG,
> 5′CATGAGCTTGAGATGCTCTCTGGGACAC and
> 5′GGCGATGCCTGCTTGCCGAATATCATGG.

All mice were housed in specific pathogen-free conditions in the CNB-CSIC animal house. Animal procedures were performed in accordance with national and EU guidelines, with the approval of the Centro Nacional de Biotecnología Animal Ethics Committee, CSIC and Comunidad de Madrid (Reference: PROEX 316/15 and PROEX 071/19).

### Antibody

The description of all antibodies and dilution used in this study is provided in *Table 2*.

## Cell culture, stimulation, transfection, and lysis

MEFs were cultured as described previously (*Sabio et al., 2005*). MEF was incubated in Dulbecco's Modified Eagle Medium (DMEM) overnight in the absence of serum before stimulation with 0.5 M sorbitol, then lysed in lysis buffer ([50 mM Tris–HCl (pH 7.5), 1 mM Ethylene glycol-bis(β-aminoethyl ether)-N,N,N′,N′-tetraacetic acid tetrasodium salt (EGTA), 1 mM Ethylenediaminetetraacetic acid (EDTA), 50 mM sodium fluoride, 10 mM sodium β-glycerophosphate, 5 mM pyrophosphate, 0.27 M sucrose, 1% (vol/vol) Triton X-100] plus 0.1% (vol/vol) 2-mercaptoethanol, 0.1 mM phenylmethylsulfonyl fluoride, 1 mM benzamidine, and 1 mM sodium orthovanadate). Lysates were centrifuged at 20,800 × *g* for 15 min at 4°C, the supernatants removed, quick frozen in liquid nitrogen and stored at −80°C until used. Transfection of MEF was carried out using Lipofectamine 2000 following the manufacture's instructions. Plasmid used were pRL-Null (Promega), 3xMEF2 Luc (Addgene), and pCMV5-Flag MEF2D (Division of Signal Transduction Therapy, University of Dundee). pCMV5-Flag MEF2D was mutated at site Ser444 (MEF2D$^{S444A}$) by directed mutagenesis method using the NZY Mutagenesis kit (NZYTech) following the manufacturer's instructions. The primer used were: *Forward*: 5′ TCA CGG CTT GGG GCC ACC GGT TCT GAC 3′; *Revers*: 5′ GTC AGA ACC GGT GGC CCC AAG CCG TGA 3′. Cells were lysed 24–30 hr after transfection.

BMDMs were isolated from adult mouse femur and tibia as described elsewhere (*Risco et al., 2012*). BMDMs were stimulated in 0.1–1% serum with 100 ng/ml LPS (Sigma-Aldrich) and lysed.

Peritoneal macrophages were isolated by peritoneal lavage with sterile and cold phosphate-buffered saline, 3 days after peritoneal injection with 2 ml of 3% (wt/vol) thioglycolate. Cells were plated in RPMI 1640 medium supplemented with 10% (vol/vol) foetal bovine serum (FBS), 2 mM glutamine, 100 U/ml penicillin, and 100 µg/ml streptomycin for 5 hr. To reduce basal phosphorylation, medium was changed to fresh medium containing 1% (vol/vol) FBS for 12 hr before stimulation with LPS (100 ng/ml) in fresh medium and then lysed.

For mRNA expression analysis, cells were lysed with NZYol (NZYtech) and RNA extracted using a standard protocol with chloroform–isopropanol–ethanol.

## Immunoblot analysis

Protein samples were resolved in sodium dodecyl sulfate–polyacrylamide gel electrophoresis (SDS–PAGE) and transferred to nitrocellulose membranes, blocked (30 min) in TBST buffer (50 mM Tris/HCl

**Table 3.** Primer sequences used for gene expression.

| Gene | Forward (5′–3′) | Reverse (5′–3′) |
| --- | --- | --- |
| *Tnfa* | TTGAGATCCATGCCGTTG | CTGTAGCCCACGTCGTAGC |
| *Il1b* | TGGTGTGTGACGTTCCCATT | CAGCACGAGGCTTTTTTGTTG |
| *Il6* | GAGGATACCACTCCCAACAGACC | AAGTGCATCATCGTTGTTCATACA |
| *Ifnb1* | TCAGAATGAGTGGTGGTTGC | GACCTTTCAAATGCAGTAGATTCA |
| *Nos2* | CAGCTGGGCTGTACAAACCTT | CATTGGAAGTGAAGCGTTTCG |
| *Il12a* | CCACCCTTGCCCTCCTAAA | GGCAGCTCCCTCTTGTTGTG |
| *Ccl5* | ATATGGCTCGGACACCACTC | TTCTTCGAGTGACAAACACG |
| *Cxcl9* | TGCACGATGCTCCTGCA | AGGTCTTTGAGGGATTTGTAGTGG |
| *Cxcl10* | GGATCCCTCTCGCAAGGA | ATCGTGGCAATGATCTCAACA |
| *Cxcl1* | CCTTGACCCTGAAGCTCCCT | CGGTGCCATCAGAGCAGTCT |
| *Ccl2* | TCTGGGCCTGCTGTTCACA | TTGGGATCATCTTGCTGGTG |
| *Cxcl2* | CCTGGTTCAGAAAATCATCCA | CTTCCGTTGAGGGACAGC |
| *Jun* | GCACATCACCACTACACCGA | GGGAAGCGTGTTCTGGCTTAT |
| *Cd14* | CATTTGCATCCTCCTGGTTTCTGA | GAGTGGTTTTCCCCTTCCGTGTG |
| *Hdac7* | CGCAGCCAGTGTGAGTGTCT | AGTGGGTTCGTGCCGTAGAG |
| *Actb* | AAGGAGATTACTTGCTCTGGCTCCTA | ACTCATCGTACTCCTGCTTGCTGAT |

pH 7.5, 0.15 M NaCl, 0.05% (vol/vol) Tween) with 5% (wt/vol) dry milk, then incubated in TBST buffer with 5% (wt/vol) dry milk and 0.5–1 µg/ml antibody (2 hr, room temperature [RT] or overnight, 4°C). Protein was detected using fluorescently labelled secondary antibodies (Invitrogen) and the Licor Odyssey infrared imaging system.

## Immunoprecipitation

Extracts from MEF were incubated with 2 µg anti-hDlg or 2 µg anti-p38γ antibody coupled to protein-G-Sepharose. After incubation for 2 hr at 4°C, the captured proteins were centrifuged at 20,800 × $g$ for 1 min, the supernatants discarded and the beads washed twice in lysis buffer containing 0.5 M NaCl, then twice with lysis buffer alone.

## RNA-sequencing

RNA was isolated from $1 \times 10^6$ BMDM from WT or p38γ/δKIKO male mice using the RNeasy kit (QIAGEN). Biological replicate libraries were prepared using the TruSeq RNA library prep kit (Illumina) and were single-end sequenced on the Illumina HiSeq 2500 platform as described in *Blair et al., 2022*.

## Gene expression analysis

cDNA for real-time qPCR was generated from total RNA using the High Capacity cDNA Reverse Transcription Kit (Applied Biosystems). Real-time qPCR reactions were performed in triplicate as described (*Risco et al., 2012*) in MicroAmp Optical 384-well plates (Applied Biosystems). PCR reactions were carried out in an ABI PRISM 7900HT (Applied Biosystems) and SDS v2.2 software was used to analyse results by the Comparative Ct Method (ΔΔCt). X-fold change in mRNA expression was quantified relative to non-stimulated WT cells, and β-actin mRNA was used as control. Primers sequences are listed in *Table 3*.

## Cytokine array

Macrophages' culture media were collected 6 hr after LPS stimulation and kept at −20°C. 450 µl of cell culture media was treated and incubated with the Proteome profiler mouse cytokine array membrane (R&D Systems) according to the manufacturer's instructions. Proteins were visualized using chemiluminescent detection reagent and signal intensity was determined using ImageJ software.

## Phosphoproteomic analysis

Following determination of protein by Bradford, 0.5 mg peritoneal macrophage protein lysate (samples were in triplicate) were reduced and alkylated in 50 mM ammonium bicarbonate, 10 mM TCEP (Tris(2-carboxyethyl)phosphine hydrochloride; Thermo Fisher Scientific), 55 mM chloroacetamide(Sigma-Aldrich) for 1 hr at 37°C. Samples were digested with 10 µg of trypsin for 8 hr at 37°C and dried down in a speed-vac. For phosphopeptide enrichment on titanium oxide (TiO$_2$) columns (GL Sciences), tryptic peptides were dissolved in 0.25 M lactic acid/3% trifluoracetic acid (TFA)/70% acetonitrile (ACN) and centrifuged at 13,000 rpm for 5 min at RT. The supernatant was loaded on the TiO$_2$ microcolumn previously equilibrated with 100 µl of 3% TFA/70% ACN. Each microcolumn was washed with 100 µl of lactic acid solution and 200 µl of 3% TFA/70% ACN. Phosphopeptides were eluted with 200 µl of 1% NH$_4$OH pH 10 in water and acidified with 7 µl of TFA. Eluates were desalted using Oasis HLB cartridges, dried down and resolubilized in 5% ACN–0.2% formic acid (FA). The phosphopeptides were separated on a home-made reversed-phase column (150 µm i.d. by 150 mm) with a 240-min gradient from 10 to 30% ACN–0.2% FA and a 600-nl/min flow rate on a Nano-LC-Ultra-2D (Eksigent, Dublin, CA) connected to a Q-Exactive Plus (Thermo Fisher Scientific, San Jose, CA). Each full MS spectrum acquired at a resolution of 70,000 was followed by 12 tandem MS (MS–MS) spectra on the most abundant multiply charged precursor ions. Tandem MS experiments were performed using higher-energy collisional dissociation (HCD) at a collision energy of 25%. The data were processed using PEAKS 8.5 (Bioinformatics Solutions, Waterloo, ON) and a Uniprot mouse database. Mass tolerances on precursor and fragment ions were 10 ppm and 0.01 Da, respectively. Variable selected posttranslational modifications were carbamidomethyl (C), oxidation (M), deamidation (NQ), acetyl (N-term), and phosphorylation (STY). Data were normalized with total ion current. Data were processed with Perseus 1.6.3.0. Samples with less than 3 valid values in at least one condition were filtered out. Imputation was performed on missing values by replacing them with random numbers taken from the lower end

of the normal distribution (Perseus default values were used width 0.3 and shift 1.8). Two sample *t*-tests were performed on sample group pairs to get the volcano plots.

## p38MAPK in vitro kinase assay

Kinase assay were performed in 30 μl final phosphorylation reaction mixture containing GST-MEF2D (0.4 μM, 1 μg), active p38MAPK (0.5 U/ml) and 50 mM Tris–HCl pH 7.5, 0.1 mM EGTA, 10 mM MgCl$_2$, and 0.1 mM [γ$^{32}$P]ATP (specific activity: ~3 × 10$^6$ cpms). The reactions were carried out at 30°C for 60 min and terminated by adding 4× SDS–PAGE sample buffer containing 1% (vol/vol) 2-mercaptoethanol. MBP phosphorylation was performed under the same conditions, but reaction was stopped by spotting the phosphorylation reaction mixture onto P81 filtermats, washed four times in 75 mM phosphoric acid to remove ATP, washed once in acetone, dried and counted for radioactivity.

## In-gel sample digestion for LC/MS–MS

Purified recombinant GST-MEF2D was phosphorylated using active p38α or p38δ (0.5 U/ml) in a buffer containing 50 mM Tris–HCl pH 7.5, 0.1 mM EGTA, 10 mM MgCl$_2$, and 0.1 mM ATP for 1 hr at 30°C. The reaction was stopped by addition of 4× SDS–PAGE sample buffer containing 1% (vol/vol) 2-mercaptoethanol and samples were subjected to electrophoresis on 10% SDS–PAGE gels. The gels were Coomassie stained and bands corresponding to MEF2D were excised, cut into pieces (~1 mm$^2$) and manually processed for in-gel digestion. The digestion protocol was described in *Shevchenko et al., 1996* with minor variations: gel pieces were washed with 50 mM ammonium bicarbonate and then with ACN, prior to reduction (10 mM Dithiothreitol (DTT) in 25 mM ammonium bicarbonate solution) and alkylation (55 mM iodoacetamide in 50 mM ammonium bicarbonate solution). Gel pieces were then washed with 50 mM ammonium bicarbonate, with ACN (5 min each), and then were dried under a stream of nitrogen. Pierce MS-grade trypsin (Thermo Fisher Scientific) was added at a final concentration of 20 ng/μl in 50 mM ammonium bicarbonate solution, for overnight at 37°C. Peptides were recovered in 50% ACN/1% FA, dried in speed-Vac and kept at −20°C until phosphopeptide enrichment. Phosphopeptide enrichment procedure concatenated two in-house packed microcolumns, Immobilized Metal Affinity Chromatography (IMAC) and Oligo R3 polymeric reversed-phase that provided selective purification and sample desalting prior to liquid chromatography–mass spectrometry (LC–MS)/MS analysis, and was performed as previously reported (*Navajas et al., 2011*).

## Protein identification by tandem mass spectrometry (LC–MS/MS Exploris 240)

The peptide samples were analysed on a nano liquid chromatography system (Ultimate 3000 nano HPLC system, Thermo Fisher Scientific) coupled to an Orbitrap Exploris 240 mass spectrometer (Thermo Fisher Scientific). Samples (5 μl) were injected on a C18 PepMap trap column (5 μm, 100 μm I.D. × 2 cm, Thermo Scientific) at 10 μl/min, in 0.1% FA in water, and the trap column was switched on-line to a C18 PepMap Easy-spray analytical column (2 μm, 100 Å, 75 μm I.D. × 50 cm, Thermo Scientific). Equilibration was done in mobile phase A (0.1% FA in water), and peptide elution was achieved in a 30-min gradient from 4 to 50% B (0.1% FA in 80% ACN) at 250 nl/min. Data acquisition was performed using a data-dependent top 15 method, in full scan positive mode (range of 350–1200 *m/z*). Survey scans were acquired at a resolution of 60,000 at *m/z* 200, with Normalized Automatic Gain Control (AGC) target of 300% and a maximum injection time (IT) of 45 ms. The top 15 most intense ions from each MS1 scan were selected and fragmented by HCD of 28. Resolution for HCD spectra was set to 15,000 at *m/z* 200, with AGC target of 75% and maximum ion injection time of 80 ms. Precursor ions with single, unassigned, or six and higher charge states from fragmentation selection were excluded.

MS and MS/MS raw data were translated to mascot general file (mgf) format using Proteome Discoverer (PD) version 2.4 (Thermo Fisher Scientific) and searched using an in-house Mascot Server v. 2.7 (Matrix Science, London, UK) against a human database (reference proteome from Uniprot Knowledgebase). Search parameters considered fixed carbamidomethyl modification of cysteine, and the following variable modifications: methionine oxidation, phosphorylation of serine/threonine/tyrosine, deamidation of asparagine/glutamine, and acetylation of the protein N-terminus. Peptide mass tolerance was set to 10 ppm and 0.02 Da, in MS and MS/MS mode, respectively, and three missed cleavages were allowed. The Mascot confidence interval for protein identification was set to ≥95%

(p < 0.05) and only peptides with a significant individual ion score of at least 30 were considered. If there were two or more residues susceptible to phosphorylation in such a way that an alternative fragment ion assignment was possible, Mascot provided information on the probability percentage that the considered residue is phosphorylated (site analysis), so that percentages above 80% usually correspond to a highly reliable assignment of the phosphorylation site. In addition, manual validation of the identified phosphopeptides was performed.

## LPS/D-Gal-induced endotoxic shock

A combination of *E. coli* 0111:B LPS (50 µg/kg body weight) (Sigma-Aldrich) and D-Gal (1 g/kg body weight) (Sigma-Aldrich) was simultaneously injected intraperitoneally in 12-week-old WT, p38γ/δKO, and p38γ/δKIKO mice of both sexes (*Risco et al., 2012*). Plasma samples were collected 2 or 6 hr after LPS/D-Gal injection for cytokine or transaminase analysis, respectively. For TUNEL and liver haemorrhage analysis livers were collected 6 hr after LPS/D-Gal injection.

## Serum analysis

Cytokine concentrations in mouse serum samples were measured using the Luminex-based MilliPlex Mouse cytokine/chemokine immunoassay and the Luminex-based Bio-Plex Mouse Grp I Cytokine 23-Plex Panel (Bio-Rad). Transaminase activities were measured using the ALT and AST Reagent Kit (Biosystems Reagents) with a Ultrospec 3100 pro UV/Visible Spectrophotometer (Amersham Biosciences).

## Histological analysis

Histological analysis was performed on haematoxylin and eosin (H&E)-stained sections of formalin-fixed, paraffin-embedded liver.

Apoptosis was determined by IHC in deparaffinized sections using TUNEL (terminal deoxynucleotidyl transferase-mediated dUTP nick end labelling) staining. TUNEL-positive cells were counted on slides from 25 random sections per mouse. Slides were mounted for fluorescence with Hoechst-containing mounting medium (Sigma) and analysed with a TCS SP5 Microscope (Leica).

## Infection by *C. albicans*

*C. albicans* (strain SC5314) was grown on YPD Agar (Sigma Y1500) plates at 30°C for 48 hr. Eight- to twelve-week-old female mice were infected intravenously with $1 \times 10^5$ colony-forming units (CFU) of *C. albicans*. Kidney fungal burden was determined at 3 days post-infection by plating the kidney homogenates in serial dilutions on YPD agar plates (*Alsina-Beauchamp et al., 2018*).

## Luciferase reporter assay

WT, p38γ/δKO, or p38γ/δKIKO MEFs were seeded in p24 plates (25,000 cells per well) and incubated overnight. The following days, 100 ng (per well) of Flag-MEF2D or Flag-MEF2D$^{S444A}$ were co-transfected with 500 ng (per well) of luciferase plasmids MEF-luciferase and renilla, using Lipofectamine 2000 (Thermo Fisher) according to the manufacturer's instructions. After 24 hr, cells were lysed using Passive Lysis Buffer (Promega). When p38MAPK inhibitors were used, cells were treated with the indicated concentrations of BIRB796 or SB203580 6 hr prior to cell lysis. Cell lysates were used to measure Luciferase activities with the Dual-Luciferase Reporter Assay System (Promega) as indicated by the manufacturer's instructions, using an Infinite M200 luminometer (TECAN). Renilla and Firefly luciferase intensities were used to calculate luciferase activity.

## Cell recruitment analysis by flow cytometry

Leucocyte infiltration in the kidney was analysed by flow cytometry as described (*Barrio et al., 2020*). Briefly, flow cytometry analysis was performed on cell suspensions from kidney homogenates that were digested with 0.2 mg/ml liberase (Roche) and 0.1 mg/ml DNase I (Roche) for 20 min at 37°C on a shaking platform and filtered through 40 µm cell strainers (Falcon). Cells were stained with combinations of fluorescent-labelled antibodies against the cell surface markers CD45, Ly6G, and F4/80 and analysed in a Cytomics FC500 flow cytometer (Beckman Coulter). Profiles were analysed with Kaluza software (Beckman Coulter); leucocytes were gated as CD45$^+$ cells as described.

Cell subpopulations in BM and spleen were analysed as described in *Barrio et al., 2020*.

## Statistical analysis

In vitro experiments have been performed at least twice with three independent replicates per experiment. For the analysis of mouse survival, production of inflammatory molecules and transaminases, the groups' size was established according to the Spanish ethical legislation for animal experiments. At least 5 mice per group were used. Differences in mouse survival were analysed by two-way analysis of variance using GraphPad Prism software. Other data were analysed using Student's $t$-test. In all cases, p-values <0.05 were considered significant. Data are shown as mean ± standard error of the mean.

The mass spectrometry proteomics data have been deposited to the ProteomeXchange Consortium via the PRIDE (*Perez-Riverol et al., 2022*) partner repository with the dataset identifier PXD042626 and 10.6019/PXD042626. The RNA-Seq data have been deposited to the Gene Expression Omnibus (GEO) repository with the accession number GSE234776.

## Acknowledgements

We thank R Gómez-Caro, V Marquez, and V Alonso (CNB-CSIC) for technical support; the antibody purification teams (Division of Signal Transduction Therapy, University of Dundee), coordinated by H McLauchlan and J Hastie, for generation and purification of antibodies. We thank the Histology Facility at CNB-CSIC for the histological processing of biological samples. This research was funded by the MCIN/AEI/10.13039/501100011033 (PID2019-108349RB-100 and SAF2016-79792R) to AC and JJSE; and by a grant from the Francis Crick Institute, which receives its core funding from Cancer Research UK (FC001103), the UK Medical Research Council (FC001103), and the Wellcome Trust (FC001103) to SCL. AE, PF, and DG-R receive MCIN FPI fellowships, ED-M MEFP FPU fellowship, and J M-G a MEFP FPU and a MCIN-Residencia de Estudiantes fellowship. MEF2D phosphorylation sites were identified in the proteomics facility of Centro Nacional de Biotecnología (CSIC) by R Navajas. PF, DG-R, ED-M, and J M-G are in the PhD Programme in Molecular Bioscience, Doctoral School, Universidad Autónoma de Madrid, 28049-Madrid, Spain.

## Additional information

### Competing interests

Nahum Sonenberg: Reviewing editor, *eLife*. The other authors declare that no competing interests exist.

### Funding

| Funder | Grant reference number | Author |
| --- | --- | --- |
| Ministerio de Ciencia e Innovación | PID2019-108349RB-100 | Juan José Sanz-Ezquerro Ana Cuenda |
| Ministerio de Ciencia e Innovación | SAF2016-79792R | Juan José Sanz-Ezquerro Ana Cuenda |
| Francis Crick Institute | | Steven C Ley |

The funders had no role in study design, data collection and interpretation, or the decision to submit the work for publication. For the purpose of Open Access, the authors have applied a CC BY public copyright license to any Author Accepted Manuscript version arising from this submission.

### Author contributions

Alejandra Escós, Conceptualization, Software, Formal analysis, Validation, Investigation, Visualization, Methodology; Ester Diaz-Mora, Data curation, Formal analysis, Validation, Investigation, Methodology; Michael Pattison, Éric Bonneil, Resources, Software, Methodology; Pilar Fajardo, Diego González-Romero, Ana Risco, Data curation, Formal analysis, Validation, Investigation; José Martín-Gómez, Formal analysis, Validation, Investigation; Nahum Sonenberg, Resources; Seyed Mehdi Jafarnejad, Resources, Formal analysis, Methodology; Juan José Sanz-Ezquerro, Formal analysis,

Validation; Steven C Ley, Resources, Supervision, Validation, Writing - review and editing; Ana Cuenda, Conceptualization, Resources, Formal analysis, Supervision, Funding acquisition, Validation, Investigation, Visualization, Methodology, Writing - original draft, Project administration, Writing - review and editing

## Author ORCIDs
Alejandra Escós  http://orcid.org/0000-0002-2990-7920
Ester Diaz-Mora  http://orcid.org/0000-0002-4233-0943
Pilar Fajardo  http://orcid.org/0000-0001-8398-7354
Diego González-Romero  http://orcid.org/0000-0002-0564-3326
Nahum Sonenberg  http://orcid.org/0000-0002-4707-8759
Seyed Mehdi Jafarnejad  http://orcid.org/0000-0002-5129-7081
Juan José Sanz-Ezquerro  http://orcid.org/0000-0002-9084-6354
Steven C Ley  http://orcid.org/0000-0001-5911-9223
Ana Cuenda  http://orcid.org/0000-0002-9013-5077

## Ethics
All mice were housed in specific pathogen-free conditions in the CNB-CSIC animal house. Animal procedures were performed in accordance with national and EU guidelines, with the approval of the Centro Nacional de Biotecnología Animal Ethics Committee, CSIC, and Comunidad de Madrid (Reference: PROEX 316/15 and PROEX 071/19).

## Decision letter and Author response
Decision letter https://doi.org/10.7554/eLife.86200.sa1
Author response https://doi.org/10.7554/eLife.86200.sa2

## Additional files

### Supplementary files
• MDAR checklist

### Data availability
All data generated or analysed during this study are included in the manuscript and supporting file.

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
