## [Editor Report]

The authors describe a new mouse model that can be used to examine p38γ/δ MAP kinase function. The data presented are solid and convincing. The authors show that p38γ/δ MAP kinase signaling contributes to macrophage responses to endotoxin. Moreover, the authors identify Ser444 as an inhibitory site of MEF2D phosphorylation by p38d.

---

## [Decision Letter]

**Decision letter after peer review:**

Thank you for submitting your article "p38γ and p38δ modulate innate immune response by regulating MEF2D activation" for consideration by *eLife*. Your article has been reviewed by 3 peer reviewers, one of whom is a member of our Board of Reviewing Editors, and the evaluation has been overseen by Tadatsugu Taniguchi as the Senior Editor. The reviewers have opted to remain anonymous.

Essential revisions:

1) Figure 5B-D. The plasmids used for these studies are not described in the methods section.

2) Figure 5B-D. No controls for cells transfected without MEF2D are presented. What part of the detected luciferase is dependent on the transfected MEF2D?

3) Figure 5B-D. No controls for the level of transfected MEF2D expression. Is the *MEF2* expression under each condition the same?

4) Page 17. The genotyping methods for the knock-in mice are cited as Refs 14 and 29. ref 14 does include the genotyping protocol for the KO mice. However, I was unable to find a description of the genotyping protocol for the knock-in mice in Ref 29.

5) The methods used for mass spec quantitation, normalization, and statistics are unclear. More information needs to be included in the methods description.

6) The mass spec raw data should be deposited in an accessible database. Data summaries should be presented as spreadsheet appendices.

7) Significance statement. line 5. "new player to the reduced number of known" should be re-worded.

8) The text is presented as "Results and Discussion" followed by "Conclusions". These sections should be relabeled as "Results" and "Discussions".

*Reviewer #1 (Recommendations for the authors):*

This is an important study that establishes a loss-of-function model for studies of p38g/d signaling. Data are presented to show that p38g/d signaling plays a role in macrophages. Phosphoproteomics data identify changes in protein phosphorylation. MEF2D is identified as a p38d substrate. Functional studies confirm that this phosphorylation is inhibitory, as noted previously (Ref 25) – the mechanism has been previously reported as due to phosphorylation-regulated sumoylation (PMID: 16356933; not cited). This should be made clearer to the reader.

1) Figure 5B-D. The plasmids used for these studies are not described in the methods section.

2) Figure 5B-D. No controls for cells transfected without MEF2D are presented. What part of the detected luciferase is dependent on the transfected MEF2D?

3) Figure 5B-D. No controls for the level of transfected MEF2D expression. Is the *MEF2* expression under each condition the same?

4) Page 17. The genotyping methods for the knock-in mice are cited as Refs 14 and 29. ref 14 does include the genotyping protocol for the KO mice. However, I was unable to find a description of the genotyping protocol for the knock-in mice in Ref 29.

5) The methods used for mass spec quantitation, normalization, and statistics are unclear. More information needs to be included in the methods description.

6) The mass spec raw data should be deposited in an accessible database. Data summaries should be presented as spreadsheet appendices.

7) Significance statement. line 5. "new player to the reduced number of known" should be re-worded.

8) The text is presented as "Results and Discussion" followed by "Conclusions". These sections should be relabeled as "Results" and "Discussions".

*Reviewer #2 (Recommendations for the authors):*

1. Gene expression analyses revealed that p38γ/p38δ regulated the expression of numerous genes implicated in the innate immune response. However, the lack of consistent results, not only between the two different mouse models of sepsis but also between the in vivo and in vitro models of LPS stimulation, decreases the impact of the results. For example, the level of the CCL2 transcript is significantly lower in the kidney of p38γ/δKIKO mice compared with that of the WT after fungal infection (Figure 2D). In contrast, p38γ/δKIKO macrophages stimulated with LPS produced significantly more CCL2 than WT macrophages (Figure S3D). The IL6 mRNA level is significantly lower in the kidney of p38γ/δKIKO mice after fungal infection compared with that of the WT (Figure 2D), whereas no differences are observed in the LPS-induced septic shock model (Figure 2F). Although one p38γ/p38δ do not appear to be required for mediating the increase in IL6 mRNA expression in LPS-stimulated macrophages (Figure 3F), the production of IL6 is significantly lower in LPS-stimulated p38γ/δKIKO macrophages (Figure S3D). Moreover, there is no strong correlation between the changes in RNA levels associated with p38γ/δKIKO and the amount of cytokines/chemokines produced. Although one can argue that fungal infection, LPS-induced septic shock, and in vitro activation of macrophages by LPS, are not directly comparable, the authors should systematically analyse the same panel of inflammatory mediators (e.g. CCL2, CCL5, IL-1a, IL-1b, IL-6, IL-10, TNFa, iNOS) in these models to identify specific inflammatory mediators downstream p38γ/δ.

2. Evidence that p38γ/δ regulate the transcriptional activity of MEF2D is performed in fibroblasts (Figure 5B-E). This result must be confirmed in macrophages. In particular, the authors should show that (i) MEF2D is indeed a substrate of p38δ in macrophages and (ii) p38γ/δ-mediated phosphorylation of MEF2D at Ser444 negatively regulate expression of iNOS and IL-1β transcripts in macrophage.

3. The biological significance of p38γ/p38δ in macrophages in the context of septic shock has not been addressed. Therefore, an additional experiment should be performed to confirm that p38γ/δKIKO monocytes/macrophages isolated from the kidney, bone marrow, and/or blood after septic shock, exhibit impaired production of inflammatory mediators (e.g. CCL2, IL-1a, IL-1b, IL-6, IL-10, TNFa, iNOS) – a hypothesis supported by the in vitro data. These experiments would provide further evidence that decreased fungal burden (Figure 2A) and increased survival after LPS-induced septic shock (Figure 2E-I) is a consequence of an abnormal innate immune response caused by the functional loss of p38γ/p38δ signaling in macrophages.

*Reviewer #3 (Recommendations for the authors):*

Below there are several points that should be addressed to clarify the information presented and make the story more convincing.

1. On page 10, and several other places of the text and figure legends, when referring to changes in expression of at least 1.5 fold it is written: -1.5< logFC <1.5. If I understand it correctly, this means logFC <-1.5 for downregulated genes and logFC >1.5 for upregulated genes. Then shouldn´t the correct formulation be -1.5> logFC >1.5?

2. Please explain in the text or figure legends the difference between the data presented in Figure 3C and in FigS3A.

3. The enrichment analysis of Figure 3E was done using the differentially expressed genes in LPS-activated p38g/dKIKO macrophages, according to the legend, but it is not clear if this refers to p38g/dKIKO treated with LPS vs untreated, or excluding genes that are common with LPS-treated WT, i.e. 95 or 53 genes in Figure 3C, LPS 30 min Down? Please clarify.

4. Figure 4D – write phosphopeptides instead of peptides in the figure.

5. Figure 4E, the legend mentions that kinase assays using MBP were also performed but are not presented in the figure. It would be interesting to show the MBP assays together with the GST-MEF2D assays to confirm that similar kinase activity levels were used in the four samples. Based on the Coomassie, it looks like there is less protein for p38g than for the other three.

6. It is surprising that p38g only poorly phosphorylates MEF2D, given the overlapping functions reported for p38g and p38d in macrophages. It would be useful to briefly discuss the implications of this observation.

7. Figure 5B, *** is indicated in the legend but not in the figure.

8. Figure 5D, to interpret this result correctly it would be important to show that the expression levels of MEF2D WT and S444A proteins are similar.

9. Figure 5E, it is not clear how these quantifications were performed. Is there any difference in the expression levels of those genes between fibroblasts expressing MEF2D WT and S444A without LPS treatment? Are Flag-MEF2D WT and S444A expressed at similar levels?

10. Figure S4A, the blots show a clear reduction in the levels of both P-p38a and P-ERK1/2 in LPS-treated WT vs p38g/dKIKO macrophages, which could be around 50%. These changes might or might not contribute to the phospho-proteome differences reported in figure 4 between the two samples, but it would be informative to present a histogram with the quantification of the two experiments performed.

11. There are some discrepancies between the information in Figure S5B and in Table 1. According to S5B: S121 is similarly phosphorylated in control and p38a, S180, and S231 are not phosphorylated in control. S98 seems also poorly phosphorylated in control. Data for S192 is not included in S5B.

To clarify the message, I suggest using the following three signs instead of x in Table 1: – no or poor phosphorylation detected, + phosphorylation detected, and ++ stronger phosphorylation detected.

---

## [Author Response]

Essential revisions:1) Figure 5B-D. The plasmids used for these studies are not described in the methods section.

We have now included in the Materials and Method section the description of the plasmids used in Figure 5B-D.

2) Figure 5B-D. No controls for cells transfected without MEF2D are presented. What part of the detected luciferase is dependent on the transfected MEF2D?

We think that the point made by the reviewers is important and we have now included those results in new Figure 5B. We observe a marked increase in the luciferase activity in p38γ/δ-/- and p38γ/δKIKO cells compared to WT cells, in both cells transfected with MEF2D or non-transfected. In addition, we found a significant increase in the relative luciferase activity when cells were transfected with MEF2D compared to non-transfected cells (regardless of the genotype).

3) Figure 5B-D. No controls for the level of transfected MEF2D expression. Is the MEF2 expression under each condition the same?

Western blots showing the level of transfected MEF2D expression are now shown in Figure 5B-D. We found that MEF2D expression is similar under all conditions (Figure 5B-D).

4) Page 17. The genotyping methods for the knock-in mice are cited as Refs 14 and 29. ref 14 does include the genotyping protocol for the KO mice. However, I was unable to find a description of the genotyping protocol for the knock-in mice in Ref 29.

We have now described the genotyping for the p38γ/δKIKO mice in methods, and also include a list with the primers used for the PCR.

5) The methods used for mass spec quantitation, normalization, and statistics are unclear. More information needs to be included in the methods description.

We have now included more information, in the material and methods section, about normalization and statistics used for mass spec quantification.

6) The mass spec raw data should be deposited in an accessible database. Data summaries should be presented as spreadsheet appendices.

A summary of relevant mass spec data is presented in Figure 4—figure supplementary 2. All the raw data have been made accessible to the journal for the Editor and the reviewers’ inspection; and have been deposited in an accessible database. The following information is now on page 26:

"The mass spectrometry proteomics data have been deposited to the ProteomeXchange Consortium via the PRIDE (*36*) partner repository with the dataset identifier PXD042626 and 10.6019/PXD042626. The RNA-sequencing data have been deposited to the Gene Expression Omnibus (GEO) repository with the accession number GSE234776".

7) Significance statement. line 5. "new player to the reduced number of known" should be re-worded.

We have now re-worded the sentence on line 5 of Significant statement as follows:

“adding a new member to the small list of known p38γ/p38δ substrates”.

8) The text is presented as "Results and Discussion" followed by "Conclusions". These sections should be relabeled as "Results" and "Discussions".

We have now re-labeled the section as Results and Discussion.

Reviewer #1 (Recommendations for the authors):This is an important study that establishes a loss-of-function model for studies of p38g/d signaling. Data are presented to show that p38g/d signaling plays a role in macrophages. Phosphoproteomics data identify changes in protein phosphorylation. MEF2D is identified as a p38d substrate. Functional studies confirm that this phosphorylation is inhibitory, as noted previously (Ref 25) – the mechanism has been previously reported as due to phosphorylation-regulated sumoylation (PMID: 16356933; not cited). This should be made clearer to the reader.

We thank the reviewer her/his comment and suggestion. We have included the new reference (Gregoire et al., 2006 JBC) and mentioned in the text that Ser444 is required for Lys439 phosphorylation that regulates MEF2D transcriptional activity.

1) Figure 5B-D. The plasmids used for these studies are not described in the methods section.

We have now included in the Materials and Method section the description of the plasmids used in Figure 5B-D.

2) Figure 5B-D. No controls for cells transfected without MEF2D are presented. What part of the detected luciferase is dependent on the transfected MEF2D?

We think that the point made by the reviewers is important and we have now included those results in new Figure 5. Similar to what found in cells transfected with MEF2D, we observe a marked increase in the luciferase activity in p38γ/δ-/- and p38γ/δKIKO cells compared to WT cells. In addition, we found a significant increase in the relative luciferase activity when cells (regardless of the genotype) are transfected with MEF2D compared to non-transfected cells.

3) Figure 5B-D. No controls for the level of transfected MEF2D expression. Is the MEF2 expression under each condition the same?

Wester blots showing the level of transfected MEF2D expression are now shown in Figure 5B-D. We found that *MEF2* expression is similar under all conditions (Figure 5B-D).

4) Page 17. The genotyping methods for the knock-in mice are cited as Refs 14 and 29. ref 14 does include the genotyping protocol for the KO mice. However, I was unable to find a description of the genotyping protocol for the knock-in mice in Ref 29.

We have now described the genotyping for the p38γ/δKIKO mice in methods, and also include a list with the primers used for the PCR.

5) The methods used for mass spec quantitation, normalization, and statistics are unclear. More information needs to be included in the methods description.

We have now included more information, in the material and methods section, about normalization and statistics used for mass spec quantification.

6) The mass spec raw data should be deposited in an accessible database. Data summaries should be presented as spreadsheet appendices.

A summary of relevant mass spec data is presented in Figure 4—figure supplementary 2. All the raw data have been made accessible to the journal for the Editor and the reviewers’ inspection; and have been deposited in an accessible database. The following information is now on page 26:

"The mass spectrometry proteomics data have been deposited to the ProteomeXchange Consortium via the PRIDE (*36*) partner repository with the dataset identifier PXD042626 and 10.6019/PXD042626. The RNA-sequencing data have been deposited to the Gene Expression Omnibus (GEO) repository with the accession number GSE234776".

7) Significance statement. line 5. "new player to the reduced number of known" should be re-worded.

We have now re-worded the sentence on line 5 of Significant statement as follows:

“adding a new member to the small list of known p38γ/p38δ substrates”.

8) The text is presented as "Results and Discussion" followed by "Conclusions". These sections should be relabeled as "Results" and "Discussions".

We have now re-labeled the section as Results and Discussion.

Reviewer #2 (Recommendations for the authors):1. Gene expression analyses revealed that p38γ/p38δ regulated the expression of numerous genes implicated in the innate immune response. However, the lack of consistent results, not only between the two different mouse models of sepsis but also between the in vivo and in vitro models of LPS stimulation, decreases the impact of the results. For example, the level of the CCL2 transcript is significantly lower in the kidney of p38γ/δKIKO mice compared with that of the WT after fungal infection (Figure 2D). In contrast, p38γ/δKIKO macrophages stimulated with LPS produced significantly more CCL2 than WT macrophages (Figure S3D). The IL6 mRNA level is significantly lower in the kidney of p38γ/δKIKO mice after fungal infection compared with that of the WT (Figure 2D), whereas no differences are observed in the LPS-induced septic shock model (Figure 2F). Although one p38γ/p38δ do not appear to be required for mediating the increase in IL6 mRNA expression in LPS-stimulated macrophages (Figure 3F), the production of IL6 is significantly lower in LPS-stimulated p38γ/δKIKO macrophages (Figure S3D). Moreover, there is no strong correlation between the changes in RNA levels associated with p38γ/δKIKO and the amount of cytokines/chemokines produced. Although one can argue that fungal infection, LPS-induced septic shock, and in vitro activation of macrophages by LPS, are not directly comparable, the authors should systematically analyse the same panel of inflammatory mediators (e.g. CCL2, CCL5, IL-1a, IL-1b, IL-6, IL-10, TNFa, iNOS) in these models to identify specific inflammatory mediators downstream p38γ/δ.

As the reviewer pointed out, fungal infection, LPS-induced septic shock, and in vitro activation of macrophages by LPS, are not directly comparable, which is the main reason why differences are observed in the production of molecules such as CCL2 or IL6. Different cell populations and tissues are implicated in /contribute to the production of cytokines and chemokines in the in vivo models; in addition, we have evidence that p38γ and p38δ have different roles depending on the cellular context (unpublished results), which could also account for the differences observed. Moreover, p38γ and p38δ can regulate cytokine production at both transcriptional and posttranscriptional levels, which could explain some of the differences found between mRNA and protein expressions.

In this work, we have analysed the most relevant cytokine for each particular in vivo model (trying to minimize the number of mice used). In cells, we have studied many more as it is shown in the figures; however, we have focused on the expression of the ones we consider more relevant for this study.

2. Evidence that p38γ/δ regulate the transcriptional activity of MEF2D is performed in fibroblasts (Figure 5B-E). This result must be confirmed in macrophages. In particular, the authors should show that (i) MEF2D is indeed a substrate of p38δ in macrophages and (ii) p38γ/δ-mediated phosphorylation of MEF2D at Ser444 negatively regulate expression of iNOS and IL-1β transcripts in macrophage.

We agree with the reviewer that the confirmation in macrophages of our results in fibroblasts would be interesting; however, we found that to express mutant MEF2D-S444A in these cells is not only technically very complicated, but also produced non-reproducible results, since macrophage activation is affected by the transfection. Nonetheless, we think that we have strong evidence suggesting that p38γ/p38δ regulate MEF2D in macrophages based on:

(1) phospho-proteomic data in macrophages showing the MEF2D Ser444 phospho-peptide,

(2) in vitro kinase data showing that MEF2D was phosphorylated at Ser444 by p38γ/p38δ, and

(3) mutation of MEF2D Ser444 to Ala increased its transcriptional activity and the expression of MEF2D dependent genes, similarly to what is observed in LPS-stimulated macrophages from p38γ/δKIKO mice (Figure 5A).

3. The biological significance of p38γ/p38δ in macrophages in the context of septic shock has not been addressed. Therefore, an additional experiment should be performed to confirm that p38γ/δKIKO monocytes/macrophages isolated from the kidney, bone marrow, and/or blood after septic shock, exhibit impaired production of inflammatory mediators (e.g. CCL2, IL-1a, IL-1b, IL-6, IL-10, TNFa, iNOS) – a hypothesis supported by the in vitro data. These experiments would provide further evidence that decreased fungal burden (Figure 2A) and increased survival after LPS-induced septic shock (Figure 2E-I) is a consequence of an abnormal innate immune response caused by the functional loss of p38γ/p38δ signaling in macrophages.

It is well documented that innate immune response is essential in septic shock and that particularly macrophages are important players in the pathogenesis of the disease. Monocyte/macrophages have a critical role in the production of cytokines and chemokines in sepsis (PMID: 28436424, PMID: 31530089). We have shown evidence of the biological significance of p38γ/p38δ in macrophages in the context of septic shock (in this work and in PMID: 29661910; PMID: 22733747; PMID: 36846591). In addition, using conditional knockout mice in which p38γ/p38δ have been deleted in myeloid cells, we have shown increased survival after LPS-induced septic shock (see Author response image 1) or after *Candida albicans* infection (PMID: 29661910), supporting the role of p38γ/p38δ in the innate immune response.

**Author response image 1. sa2fig1:** (Left panel) WT (*n* = 8) and LysMCre-p38γ/δ^-/-^ (*n*= 7) mice were injected with LPS (20 μg/g), and survival was monitored for up to 5 days. Graph shows % survival at the indicated times. (Right panel) 30 μg of protein lysates from WT and LysMCre-p38γ/δ^-/-^ peritoneal macrophage were immunoblotted with the indicated antibodies to total p38γ, p38δ, and tubulin as loading control. (*) unspecific band.

Reviewer #3 (Recommendations for the authors):Below there are several points that should be addressed to clarify the information presented and make the story more convincing.1. On page 10, and several other places of the text and figure legends, when referring to changes in expression of at least 1.5 fold it is written: -1.5< logFC <1.5. If I understand it correctly, this means logFC <-1.5 for downregulated genes and logFC >1.5 for upregulated genes. Then shouldn´t the correct formulation be -1.5> logFC >1.5?

We thank the reviewer for noticing the mistake. The reviewer is right and we have now corrected this in the manuscript.

2. Please explain in the text or figure legends the difference between the data presented in Figure 3C and in FigS3A.

Data Figure S3A showed the overlap of genes up- or down-regulated in p38γ/δKIKO compared with WT, over the time course, and the genes that were common between different times. After the referees comment we agree that these data are similar to the ones shown in Figure 3A-C and we have decided to delete Figure S3A from the paper. As mentioned, data in old Figure S3A were redundant with the ones shown in Figure 3 and were causing confusion without adding much information. We have now included a sentence in the text indicating that:

“Different gene profiles of p38γ/δKIKO and WT BMDM at basal conditions (time 0) indicates that p38γ and p38δ also regulate gene expression in resting conditions (Figure 3B)”.

3. The enrichment analysis of Figure 3E was done using the differentially expressed genes in LPS-activated p38g/dKIKO macrophages, according to the legend, but it is not clear if this refers to p38g/dKIKO treated with LPS vs untreated, or excluding genes that are common with LPS-treated WT, i.e. 95 or 53 genes in Figure 3C, LPS 30 min Down? Please clarify.

The enrichment analysis of Figure 3E was done using the differentially expressed genes in LPS-activated p38γ/δKIKO macrophages excluding genes that are common with LPS-treated WT. We have now clarified this in the figure legend. In addition, we have highlighted in bold the genes that are negatively regulated in old Figure S3B (new Figure S3A) as reviewer 2 asked.

4. Figure 4D – write phosphopeptides instead of peptides in the figure.

We have now written phosphopeptides instead of peptides in Figure 4D as the reviewer asked.

5. Figure 4E, the legend mentions that kinase assays using MBP were also performed but are not presented in the figure. It would be interesting to show the MBP assays together with the GST-MEF2D assays to confirm that similar kinase activity levels were used in the four samples. Based on the Coomassie, it looks like there is less protein for p38g than for the other three.

We agree with the reviewer and we have now included the data from the kinase assay using MBP as substrate. These results are now shown in new Figure 4G and the assay is described in Methods.

6. It is surprising that p38g only poorly phosphorylates MEF2D, given the overlapping functions reported for p38g and p38d in macrophages. It would be useful to briefly discuss the implications of this observation.

As the reviewer points out, p38γ is less efficient than p38δ in phosphorylating MEF2D in vitro; however, we think that this phosphorylation is not negligible (approx. 50% of that of p38δ). Nonetheless, we cannot rule out the possibility that p38γ might play a more important role in the in vivo phosphorylation of MEF2D and more experiments need to be performed to investigate this matter.

As the reviewer asked, we have now discussed this observation the Discussion section (page 16-17).

7. Figure 5B, *** is indicated in the legend but not in the figure.

As the reviewer indicated *** is now shown in new Figure 5B.

8. Figure 5D, to interpret this result correctly it would be important to show that the expression levels of MEF2D WT and S444A proteins are similar.

We think that the point made by the reviewers is important and we have now included those results in new Figure 5D. The expression levels of MEF2D WT and S444A proteins are similar.

9. Figure 5E, it is not clear how these quantifications were performed. Is there any difference in the expression levels of those genes between fibroblasts expressing MEF2D WT and S444A without LPS treatment? Are Flag-MEF2D WT and S444A expressed at similar levels?

In Figure 5E, results were normalized to β-actin RNA expression and fold induction calculated relative to that in cells expressing Flag-MEF2D WT at time 0 of LPS. We have now included data at time 0 LPS in new Figure 5E. We found that, at time 0, gene expression was increased in fibroblasts expressing MEF2D S444A.

As shown in Figure 5D Flag-MEF2D WT and S444A were expressed at similar levels in MEFs.

10. Figure S4A, the blots show a clear reduction in the levels of both P-p38a and P-ERK1/2 in LPS-treated WT vs p38g/dKIKO macrophages, which could be around 50%. These changes might or might not contribute to the phospho-proteome differences reported in figure 4 between the two samples, but it would be informative to present a histogram with the quantification of the two experiments performed.

We have quantified the P-p38a and P-ERK1/2 bands in the two experiments, as the reviewer asked. These data are now shown in new Figure S4A.

11. There are some discrepancies between the information in Figure S5B and in Table 1. According to S5B: S121 is similarly phosphorylated in control and p38a, S180, and S231 are not phosphorylated in control. S98 seems also poorly phosphorylated in control. Data for S192 is not included in S5B.To clarify the message, I suggest using the following three signs instead of x in Table 1: – no or poor phosphorylation detected, + phosphorylation detected, and ++ stronger phosphorylation detected.

We have made the changes that the reviewer suggested in Table 1.